# PEAR1 regulates expansion of activated fibroblasts and deposition of extracellular matrix in pulmonary fibrosis

Yan Geng[1,8], Lin Li[1,8], Jie Yan[1], Kevin Liu[2], Aizhen Yang[3], Lin Zhang[1], Yingzhi Shen[1], Han Gao[4], Xuefeng Wu ®[5], Imre Noth[6], Yong Huang[6], Junling Liu ®[1,7] ✉ & Xuemei Fan ®[1] ✉

Pulmonary fibrosis is a chronic interstitial lung disease that causes irreversible and progressive lung scarring and respiratory failure. Activation of fibroblasts plays a central role in the progression of pulmonary fibrosis. Here we show that platelet endothelial aggregation receptor 1 (PEAR1) in fibroblasts may serve as a target for pulmonary fibrosis therapy. *Pear1* deficiency in aged mice spontaneously causes alveolar collagens accumulation. Mesenchyme-specific *Pear1* deficiency aggravates bleomycin-induced pulmonary fibrosis, confirming that PEAR1 potentially modulates pulmonary fibrosis progression via regulation of mesenchymal cell function. Moreover, single cell and bulk tissue RNA-seq analysis of pulmonary fibroblast reveals the expansion of Activated-fibroblast cluster and enrichment of marker genes in extracellular matrix development in *Pear1⁻/⁻* fibrotic lungs. We further show that PEAR1 associates with Protein Phosphatase 1 to suppress fibrotic factors-induced intracellular signalling and fibroblast activation. Intratracheal aerosolization of monoclonal antibodies activating PEAR1 greatly ameliorates pulmonary fibrosis in both *WT* and *Pear1*-humanized mice, significantly improving their survival rate.

Pulmonary fibrosis (PF) represents chronic progressive scar formation[1] seen in a variety of diseases, including systemic sclerosis, sarcoidosis, infection, and environmental exposure[2]. But in most patients, the cause of PF is unknown and is classified as idiopathic PF (IPF)[3,4]. IPF has a poor prognosis and a course that is unpredictable, which leads to irreversible progression and eventually respiratory failure and death[5].

PF is characterized by the proliferation of fibroblasts and deposition of the extracellular matrix (ECM), leading to structural destruction of lung tissue[6]. Fibroblasts synthesize and secrete proteins to form the ECM, which maintains the normal scaffold of lung epithelium and endothelium that are involved in efficient gas exchange

and in the normal repair of damaged tissue[7]. Many pro-fibrotic factors stimulate fibroblast proliferation and induce its transformation from fibroblast to myofibroblast, leading to increased synthesis of ECM[3,8]. Although resident fibroblasts are differentiated from mesenchymal progenitor cells, many studies have shown that the pulmonary fibroblasts can also be derived from epithelial cells, macrophages, blood monocytes, endothelial cells, and other lung cells in the process of IPF[9].

PEAR1 belongs to a unique family of EGF (epidermal growth factor) repeat-containing transmembrane proteins[10], and is majorly expressed in endothelial cells and platelets[10–13]. The high-affinity

[1]Department of Biochemistry and Molecular Cell Biology, Shanghai Jiao Tong University School of Medicine, Shanghai, China. [2]College of Arts and Science, Vanderbilt University, Nashville, TN, USA. [3]Cyrus Tang Hematology Center, Soochow University, Suzhou, China. [4]State Key Laboratory of Cognitive Neuroscience and Learning and Beijing Key Laboratory of Genetic Engineering Drugs & Biotechnology, College of Life Sciences, Beijing Normal University, Beijing, China. [5]Shanghai Institute of Immunology, Department of Immunology and Microbiology, Shanghai Jiao Tong University School of Medicine, Shanghai, China. [6]Department of Medicine, University of Virginia, Charlottestville, VA, USA. [7]Shanghai Synvida Biotechnology Co., Ltd, Shanghai, China. [8]These authors contributed equally: Yan Geng, Lin Li. ✉e-mail: liujl@shsmu.edu.cn; fanxuemei@sjtu.edu.cn

immunoglobulin E receptor α (FcεR1α) has been identified as a natural ligand for human PEAR1. FcεR1α potentiates platelet aggregation and leads to PEAR1 phosphorylation by directly binding to EGF-like repeat 13 of PEAR1 in human platelets[14]. PEAR1 has also been reported to be a signalling receptor for dextran sulfate and fucoidan in human, but not mouse, platelets[15]. However, a study has also shown that *Pear1* deficiency does not affect murine platelet function[16]. PEAR1 has also been reported to be expressed in endothelial cells and that Pear1 deficiency may promote skin wound healing by promoting neoangiogenesis[17]. Therefore, PEAR1 plays diverse biological functions in different tissues and organs.

Here, we found that *Pear1* deficiency leads to increased PF in both aging and the bleomycin (bleo) mouse model. PEAR1 regulates fibroblast activation by associating with Protein Phosphatase 1 to suppress fibrotic factors-induced intracellular signaling. Importantly, monoclonal antibodies activating PEAR1 suppressed matrix protein deposition and ameliorated the degree of PF in bleo mouse model.

## Results

### *Pear1* deficiency aggravated bleo-induced PF by direct regulation of mesenchymal cells function

By observing aged mice, we found that the forced vital capacity (FVC) of the *Pear1⁻/⁻* mice were significantly lower than those of the *WT* control mice (Fig. 1a). Masson's trichrome staining and immunostaining revealed that *Pear1* deficiency caused a significant collagen accumulation in pulmonary alveoli (Fig. 1b, Supplementary Fig. 1b), which indicated that *Pear1* deficiency spontaneously caused mild PF.

Bleo-induced lung fibrosis is the most common method to induce PF in murine models[18]. Mice were intratracheally aerosolized with bleo at different concentrations, respectively. The mortality and weight loss of *Pear1⁻/⁻* mice were significantly higher than that of *WT* mice during the 21 days' observation period (Fig. 1d, Supplementary Fig. 1c). The respiratory function was further measured in the survival mice in response to the low dose bleo. The results in Fig. 1c showed that the FVC of the *Pear1⁻/⁻* mice were significantly lower than those of the *WT* control mice in response to bleo. Masson's trichrome staining and immunostaining revealed that intratracheal aerosolization of bleo caused a pulmonary consolidation in mice, and *Pear1* deficiency significantly aggravated collagen accumulation (Fig. 1e, Supplementary Fig. 1d) and increased Pdgfra⁺ mesenchymal cells (Supplementary Fig. 1d), indicating that *Pear1* deficiency may exacerbate PF by aggravating collagen accumulation and mesenchymal cells proliferation.

The proportion of endothelial cells, leukocytes, epithelial cells, and fibroblasts in the lung tissue of WT and *Pear1* deficient mice with or without treatment of bleo were further analyzed (Fig. 1f). We found that *Pear1* deficiency and bleo treatment had no effects on the proportion of pulmonary endothelial cells (CD31⁺). However, the proportion of leukocytes (CD45⁺) in mice treated with bleo was significantly increased, but *Pear1* deficiency did not affect the rising ratio of leukocytes. The ratio of epithelial cells (CD326⁺) in *WT* and *Pear1* deficient mice treated by bleo was greatly reduced, and the percentage of epithelial cells in *Pear1⁻/⁻* mice was significantly lower than that in the control group, indicating that *Pear1* deficiency aggravated bleo-induced lung epithelial injury. Bleo treatment significantly increased the proportion of Pdgfra⁺ mesenchymal cells (CD31⁻CD45⁻CD326⁻) in *Pear1* deficient mice compared to the control mice (Fig. 1f). We also found that the proportion of Pdgfra⁺ mesenchymal cells that were PEAR1 positive was three folds higher in PF *WT* mice compared to the control (Fig. 1g). Moreover, PEAR1 was expressed in almost all cultured fibroblasts isolated from *WT* lung tissue and inhibited the proliferation of cultured fibroblasts (Supplementary Fig. 1g, h). The number of ki67⁺ fibroblasts in lung sections from *Pear1⁻/⁻* mice were higher than that from *WT* mice (Supplementary

Fig. 1i, j). These results indicated that the *Pear1* deficiency promoted proliferation of mesenchymal cells.

Mesenchymal cell-specific *Pear1* knockout mice were further constructed using tamoxifen-induced *Col1a2-Cre* strategy[19] to investigate the function of PEAR1 in mesenchymal cells in PF (Supplementary Fig. 2a). After tamoxifen treatment, *Col1a2-Cre^ER^Pear1^f/f^* and control mice were intratracheally aerosolized with bleo, respectively. The mortality and weight loss of *Col1a2-Cre^ER^Pear1^f/f^* mice were significantly higher than that of control mice during the 21 days' observation period in response to bleo (Fig. 1i, Supplementary Fig. 2b). The results in Fig. 1h showed that the FVC of the *Col1a2-Cre^ER^Pear1^f/f^* mice were significantly lower than those of the control mice in response to bleo. Masson's trichrome staining and immunostaining revealed that *Pear1* deficiency significantly aggravated collagen accumulation and increased Pdgfra⁺ mesenchymal cells (Fig. 1j, Supplementary Fig. 2c, d), which further verified that *Pear1* deficiency exacerbated PF by directly regulating mesenchymal cell function.

### *Pear1* deficiency expanded activated fibroblasts in bleo model

Pulmonary mesenchymal cells (MSC, CD31⁻CD45⁻CD326⁻) were sorted by flow cytometry from fresh lung tissues as described in "Methods", and adopted for scRNA-seq assay. Data processing, quality control, and cell filtering were conducted as described in "Methods". The 17934 MSC were re-classified into eight clusters (Fig. 2a). Cell clusters under different experimental conditions were displayed in Supplementary Fig. 4. We first characterized the cell clusters with common fibroblast and MSC markers (Supplementary Fig. 5). Pdgfra⁺ cells were aggregated on the left side, including Cluster 1, 2, 5, and 8 (Supplementary Fig. 5a). Importantly, these clusters were also positive with CD34 and Inmt (Supplementary Fig. 5e, h), as well as Col14a1 or Col13a1 (Supplementary Fig. 6a, b), thus representing resident matrix fibroblast developed from bone marrow-derived progenitor cells.

Cell type annotations were carried out using a mixture of supervised approaches coupled with manual curation, as described in "Methods". A hierarchical clustering dendrogram identified four pairs of cell types plus a unique cluster of activated fibroblast[20] (Fig. 2b). The cell proportion of each cluster demonstrated that activated fibroblast was induced by bleo in *WT* (7.1%) and substantially expanded in *Pear1⁻/⁻* (15.5%), but was absent in *WT* and *Pear1⁻/⁻* mice treated by vehicle (<0.1%) (Fig. 2c). Forty-seven genes in our scRNA-seq data can be matched to the 49 genes in activated fibroblast signature[20]. GSEA analysis revealed a significant enrichment of the 39/47 genes in Cluster 7 when compared to other clusters (Fig. 2d), confirming the assignment of Cluster 7 to activated fibroblast. Notably, Cluster 7 was positive in both Pdgfra and Pdgfrb (Supplementary Fig. 5a, b).

Markers for the selected clusters are displayed in Fig. 2e. Cluster 4 cells displayed the highest level of α-smooth muscle actin (Acta2), along with other markers for fibromyocytes[21] and pericytes[22] (Fig. 2e, left and right of the blue line underneath Cluster 4, respectively). These findings indicated that Cluster 4 cells were developed from pericyte progenitor cells. Notably, while fibromyocytes in Cluster 4 were drastically induced in *WT* bleo (22.1%), its proportion was ameliorated in *Pear1⁻/⁻* bleo (14.9%) (Fig. 2c, Supplementary Fig. 4). In addition, markers for Cluster 1, 2, 3, 5, and 6 are displayed in Supplementary Fig. 6a−e, respectively. Two subclusters were identified in cluster 6 (Supplementary Fig. 7a). Markers for these two subclusters indicated that subcluster 0 was airway SMC, while subcluster 1 was lipofibroblasts (Supplementary Fig. 7b, c, respectively)[23]. These two subclusters demonstrated little difference between *Pear1⁻/⁻* bleo and *WT* bleo (Supplementary Fig. 7a).

Functional enrichment analysis of marker genes for activated fibroblast in *Pear1⁻/⁻* bleo revealed significant GO biological processes involved in extracellular organization and connective tissue development (Fig. 2f), and KEGG pathways involved in focal adhesion, lung

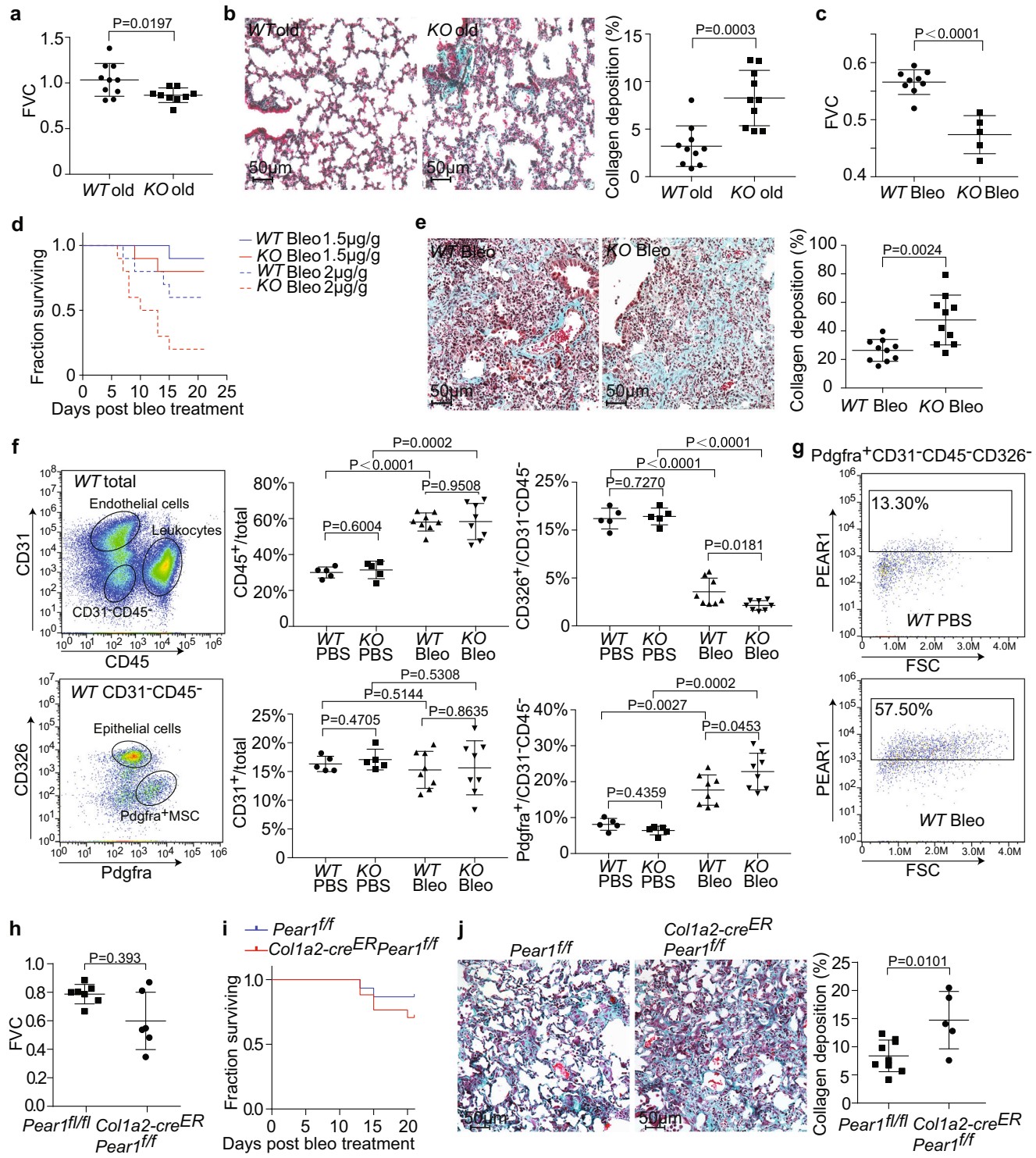

fibrosis, and TGFβ signaling (Fig. 2g). We further identified a set of differentially expressed genes (DEGs) in active fibroblast between *Pear1*[−/−] bleo and *WT* bleo (Fig. 2h). In concordance with the elevated level of α-smooth muscle actin in Fibromyocytes, markers of Cluster 4 in *Pear1*[−/−] bleo were enriched in contractile function and focal adhesion (Supplementary Fig. 8a, b). Cluster 0 was the largest cell cluster in the *WT* ctrl and *Pear1*[−/−] ctrl, but was diminished after bleo treatment (Fig. 2c, Supplementary Fig. 4). The markers of Cluster 0 in *WT* ctrl were enriched in pathways involved in cellular response to growth factor stimulus and wound healing (Supplementary Fig. 8c, d).

## PEAR1 associated with PP1 to suppress fibrotic factor induced fibroblast activation

According to single-cell phenotyping, Pdgfra[+] cells containing almost only resident matrix fibroblast and activated fibroblast were selected by flow cytometry for RNA-seq assay (Supplementary Fig. 5a). Venn diagrams of DEGs identified by two-group comparisons with criteria of fold change >2 and FDR < 0.05 are shown in Fig. 3a and Supplementary Data 1. A PCA plot showed that sample groups were separated by the treatment of bleo or vehicle at Dim-1, and by *KO* or *WT* at Dim-2 (Fig. 3b). Pathway analysis of upregulated genes in *Pear1*[−/−] bleo compared to *WT* bleo revealed significant pathways involved in ECM

**Fig. 1 | *Pear1* deficiency exacerbated PF by direct regulation of mesenchymal cells function. a** Forced vital capacity (FVC) was measured for evaluating lung function of 12-month-old *WT* mice and *Pear1*⁻ᐟ⁻ mice (*n* = 10 mice in *WT* group; *n* = 9 mice in *Pear1*⁻ᐟ⁻ group). *WT*, wild type. **b** Masson staining of lung tissue in 12-month-old *WT* mice and *Pear1*⁻ᐟ⁻ mice. The collagen area (green) was calculated (*n* = 10 mice per group) (scale bars, 50 μm). **c** FVC of *WT* and *Pear1*⁻ᐟ⁻ mice was measured on day 21 after bleo administration (*n* = 9 mice in *WT* group; *n* = 5 mice in *Pear1*⁻ᐟ⁻ group). **d** Survival curves of *WT* and *Pear1*⁻ᐟ⁻ mice induced by 1.5 μg/g or 2 μg/g bleo through endotracheal atomization (*n* = 10 mice per group). **e** Representative images of masson staining on lung sections from *WT* mice and *Pear1*⁻ᐟ⁻ mice on day 21 after bleo administration. The collagen area (green) was calculated (*n* = 10 mice per group) (scale bars, 50 μm). **f** The proportion of leukocytes (CD45⁺), endothelial cells (CD31⁺), epithelial cells (CD326⁺) and Pdgfra⁺ mesenchymal cells (MSC, CD45⁻CD31⁻CD326⁻) were analyzed in the lung tissue of *Pear1*⁻ᐟ⁻ mice with or without treatment of bleo (*n* = 5 mice in PBS group; *n* = 8 mice in bleo group). **g** PEAR1⁺ cell proportion in Pdgfra⁺ mesenchymal cells (Pdgfra⁺CD45⁻CD31⁻CD326⁻) from *WT* mice lung treated with or without bleo was investigated by flow cytometry. **h** FVC was measured of *Pear1*ᶠ/ᶠ and *Col1a2-Cre*ᴱᴿ*Pear1*ᶠ/ᶠ mice on day 21 after bleo administration (*n* = 9 mice in *Pear1*ᶠ/ᶠ group; *n* = 5 mice in *Col1a2-Cre*ᴱᴿ*Pear1*ᶠ/ᶠ group). **i** Survival curves of *Pear1*ᶠ/ᶠ and *Col1a2-Cre*ᴱᴿ*Pear1*ᶠ/ᶠ mice induced by 1.5 μg/g bleo through endotracheal atomization (*n* = 15 mice in *Pear1*ᶠ/ᶠ group; *n* = 17 mice in *Col1a2-Cre*ᴱᴿ*Pear1*ᶠ/ᶠ group). **j** Representative images of masson staining on lung sections from *Pear1*ᶠ/ᶠ and *Col1a2-Cre*ᴱᴿ*Pear1*ᶠ/ᶠ mice on day 21 after bleo administration. The collagen area (green) was calculated (*n* = 9 mice in *Pear1*ᶠ/ᶠ group; *n* = 5 mice in *Col1a2-Cre*ᴱᴿ*Pear1*ᶠ/ᶠ group) (scale bars, 50 μm). For **a**–**c**, **e**, **h**, **j**, two-tailed t test was used. For **f**, one-way ANOVA was used. Data are presented as mean ± SD. Source data are provided as a Source data file.

organization and development (Fig. 3c). GSEA of these two groups highlighted EMT for *Pear1*⁻ᐟ⁻ bleo mice (Supplementary Fig. 9a, FDR = 0.005). We performed qPCR to validate the GSEA leading-edge EMT genes for collagen synthesis and ECM development (Supplementary Fig. 9b). A heat map across all samples showed that EMT leading-edge genes in GSEA were uniquely upregulated in *Pear1*⁻ᐟ⁻ bleo, but not in *WT* bleo (Fig. 3d). Notably, these EMT genes, when mapped to single-cell RNA-seq data, demonstrated predominant expression in activated fibroblast cluster (Fig. 3e). Hence, our integrated analysis of bulk and single cells RNA-seq revealed EMT in *Pear1*⁻ᐟ⁻ bleo mice, likely carried-out by activated fibroblast.

There are several cytokines have been reported to activate fibroblast and are involved in PF, such as transforming growth factor β (TGFβ), fibroblast growth factor (FGF), and platelet-derived growth factor (PDGF)[24]. TGFβ, FGF, PDGF can activate Smad pathway, MAPKs pathway, and PI3K/Akt pathway to directly promote fibroblast activation and proliferation. To investigate the roles of PEAR1 in the regulation of the signaling pathways mediated by TGFβ, FGF and PDGF, *WT* and *Pear1*⁻ᐟ⁻ mouse pulmonary fibroblasts were stimulated with TGFβ, FGF, and PDGF, respectively. We found that the phosphorylation levels of Smad2/3, Akt, ERK1/2, P38, and JNK1/2 were significantly enhanced in *Pear1*⁻ᐟ⁻ fibroblasts in response to TGFβ, PDGF, or FGF stimulation, respectively (Fig. 3f), suggesting that both canonical and non-canonical signaling pathway can be negatively regulated by PEAR1.

The possible associated proteins of PEAR1 in fibroblast were also studied through the immunoprecipitation of mouse fibroblasts lysate with PEAR1 antibody. Protein mass spectrometry showed that cytoplasmic protein phosphatase 1 catalytic subunit alpha (PP1α) could interact with PEAR1 (Fig. 3g, h, Supplementary Fig. 9c, d). PP1α is one of the three catalytic subunits of protein phosphatase 1 (PP1). PP1 inhibitor Calyculin A was used to treat *Pear1*⁺ᐟ⁺ pulmonary fibroblasts. The results showed that PP1 inhibitor significantly promoted the phosphorylation levels of Smad2/3, P38, JNK1/2, and AKT (Fig. 3i, Supplementary Fig. 9e), and the synthesis of ECM proteins (Fig. 3j) in cultured pulmonary fibroblasts, showing that PEAR1 may associate with PP1 to inhibit the activation of fibroblasts by suppressing the phosphorylation of signaling protein. PEAR1 has been reported to be a regulator of PI3K-AKT signaling by reducing the expression of PTEN[25]. However, *Pear1* deficiency neither affected the expression of PTEN nor directly interacted with PTEN in fibroblasts (Supplementary Fig. 9f, g), which excluded the possibility that PEAR1 regulates fibroblast activation through PTEN.

### Monoclonal antibody targeted human PEAR1 for PF therapy
We found that intratracheal nebulization of monoclonal antibody (mAb) LF1 could activate mouse PEAR1 and effectively improve bleo-induced mouse PF in a PEAR1-dependent manner (Supplementary Fig. 10). mAb LF2 targeting human PEAR1 were further prepared with the ability to inhibit the synthesis of periostin (Postn) and various Collagens in human pulmonary fibroblast (Fig. 4a). To evaluate the effects of LF2 on PF in vivo, a transgenic mouse with human *Pear1* gene replacing mouse *Pear1* gene was successfully established (Supplementary Fig. 11a–c). LF2 was intratracheally aerosolized twice on the 7th and 14th day after bleo treatment. LF2 could permeate the epithelial cells and distribute in the interstitium of the lung (Supplementary Fig. 11d). In order to objectively evaluate the therapeutic effect of LF2 antibody, pirfenidone was selected as a positive drug control. Based on the literature method, human *Pear1* transgenic mice were given pirfenidone by gavage once a day at one day before and the following 20 days after bleo treatment. On the 21st day of bleo treatment, the lung function of the mice was measured and the mice were sacrificed for further pathological analysis. The results showed LF2 dose-dependently reduced the mortality of mice (Fig. 4b), improved respiratory function (Fig. 4c), and suppressed the deposition of collagen (Fig. 4d) and fibroblasts proliferation (Fig. 4e, f). Compared with pirfenidone, 1 mg/kg of LF2 had better effects on improving respiratory function, reducing ECM protein deposition and fibroblast proliferation.

The role of PEAR1 in fibrosis was then evaluated by an amiodarone-induced pulmonary fibrosis model[26,27]. The results showed that *Pear1* deficiency significantly aggravated collagen accumulation, while LF2 suppressed the deposition of ECM induced by amiodarone (Supplementary Fig. 11e, f). These data indicate that the role of PEAR1 could be reproducible in different pulmonary fibrosis models and may be a promising target for pulmonary fibrosis therapy (Fig. 4g).

Furth, the function of LF2 on normal (HFL1) and PF fibroblasts was detected. The result showed that the synthesis of ECM proteins was significantly inhibited by LF2 in a concentration-dependent manner in both HLF1 and PF fibroblasts in response to TGFβ (Supplementary Fig. 12). These data indicated that fibroblasts from established fibrotic lungs could respond to the LF2 antibody.

## Discussion
PEAR1 is expressed on the surface of vascular endothelial cells and platelets. Our results showed that PEAR1 is not involved in murine platelet activation (Supplementary Fig. 13a–e). Previous findings have shown that bleomycin-induced pulmonary fibrosis is correlated with platelet trapping and that anti-CD11a antibodies decrease platelet trapping and collagen deposition[28,29], indicating that platelets may be directly involved in the progression of Bleo-induced pulmonary fibrosis and independent of platelet activation. However, the possibility of PEAR1 regulating PF through platelets has been clearly exclude by the platelet-specific *Pear1*-deficient mice (Supplementary Fig. 13h–k). Neoangiogenesis was significantly increased in a hind limb ischemia ligation model in *Pear1* deficient mice, seen from the increased capillary density[17], although the role of angiogenesis in PF remains equivocal[30,31]. We found that *Pear1* deficiency had no effect on neither the morphology of pulmonary vessels in PF mice nor the tubulogenesis of pulmonary endothelial cells isolated from mice

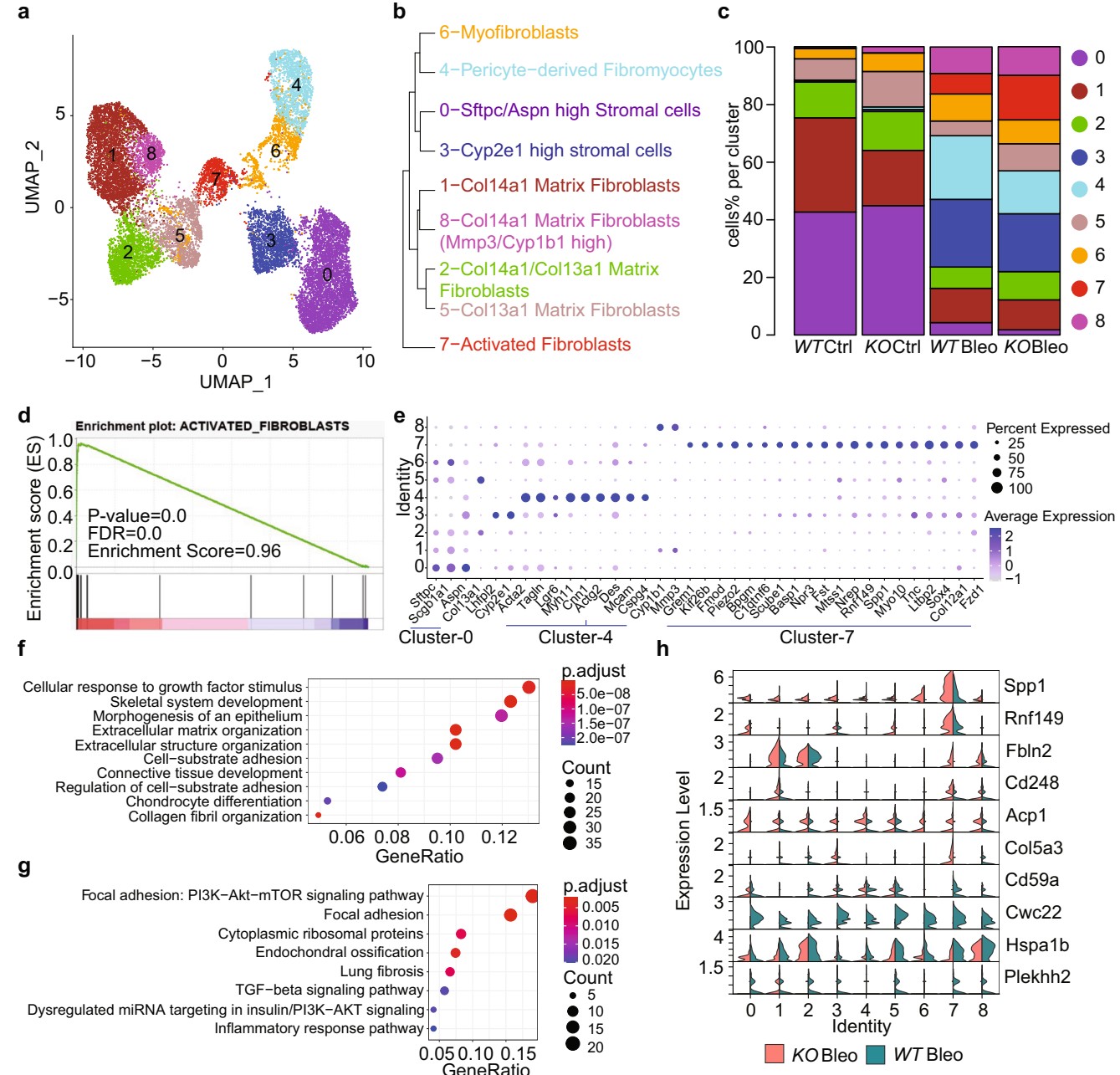

**Fig. 2 | *Pear1* deficiency expanded activated fibroblasts in bleo model. a–c** All of the 17,934 mesenchymal cells across four experiment conditions were integrated for umap clustering (**a**), cell type annotation and cluster-tree dendrogram (**b**), and fraction of each cell type in each experimental condition (**c**). **d** GSEA analysis of activated fibroblasts signature of cluster 7 compared with the rest clusters. **e** Dotplot of markers for the selected cluster. **f**, **g** Top 10 GO biological process (**f**)

and KEGG pathways (**g**) enriched with Cluster 7 markers in *Pear1*[−/−] bleo mice. **h** Differentially expressed genes in Cluster 7 between *Pear1*[−/−] bleo and *WT* bleo. For **d**, nominal *P* value for the statistical significance of the enrichment score is adjusted using Benjamini–Hochberg method for multiple gene sets and hypothesis testing. For **f**, **g**, one-sided Fisher's exact test was used.

(Supplementary Fig. 13f, g). These results exclude the possibility that PEAR1 may regulate PF through endothelial cells. Col1a2-Cre-driven mesenchymal cell-specific *Pear1* deficiency significantly exacerbated bleo-induced PF, indicating that PEAR1 is mainly involved in the process of PF through the modulation of mesenchymal cells.

ScRNA-seq analysis identified 4 clusters of fibroblasts induced or increased by bleo treatment, i.e., Cluster 3, 4, 7, and 8 (Fig. 2c, Supplementary Fig. 4). GSEA revealed significant enrichment of the 49-gene signature[20] in Cluster 7, thus defined as activated fibroblast (Fig. 2d). Accordingly, functional analysis revealed an enrichment of activated fibroblast marker genes in ECM development and lung

fibrosis (Fig. 2f, g), indicating that *Pear1* deficiency reshapes mesenchymal components to a new pattern that favors the exaggeration of the disease through higher collagen deposition, decreased pulmonary function and poor prognosis (Fig. 1). The progenitor of this cell population remains unknown, although it may be heterogeneous in origin. RNA-seq analysis of Pdgfra[+] bulk tissue further confirmed that multiple pathways involved in ECM organization and development were enriched with upregulated genes in *Pear1*[−/−] mouse fibrotic lung. Among these pathways, the genes of EMT were upregulated only in *Pear1*[−/−] Bleo but not in *WT* Bleo. Interestingly, these EMT genes were preferentially expressed in activated fibroblast in scRNA-seq data.

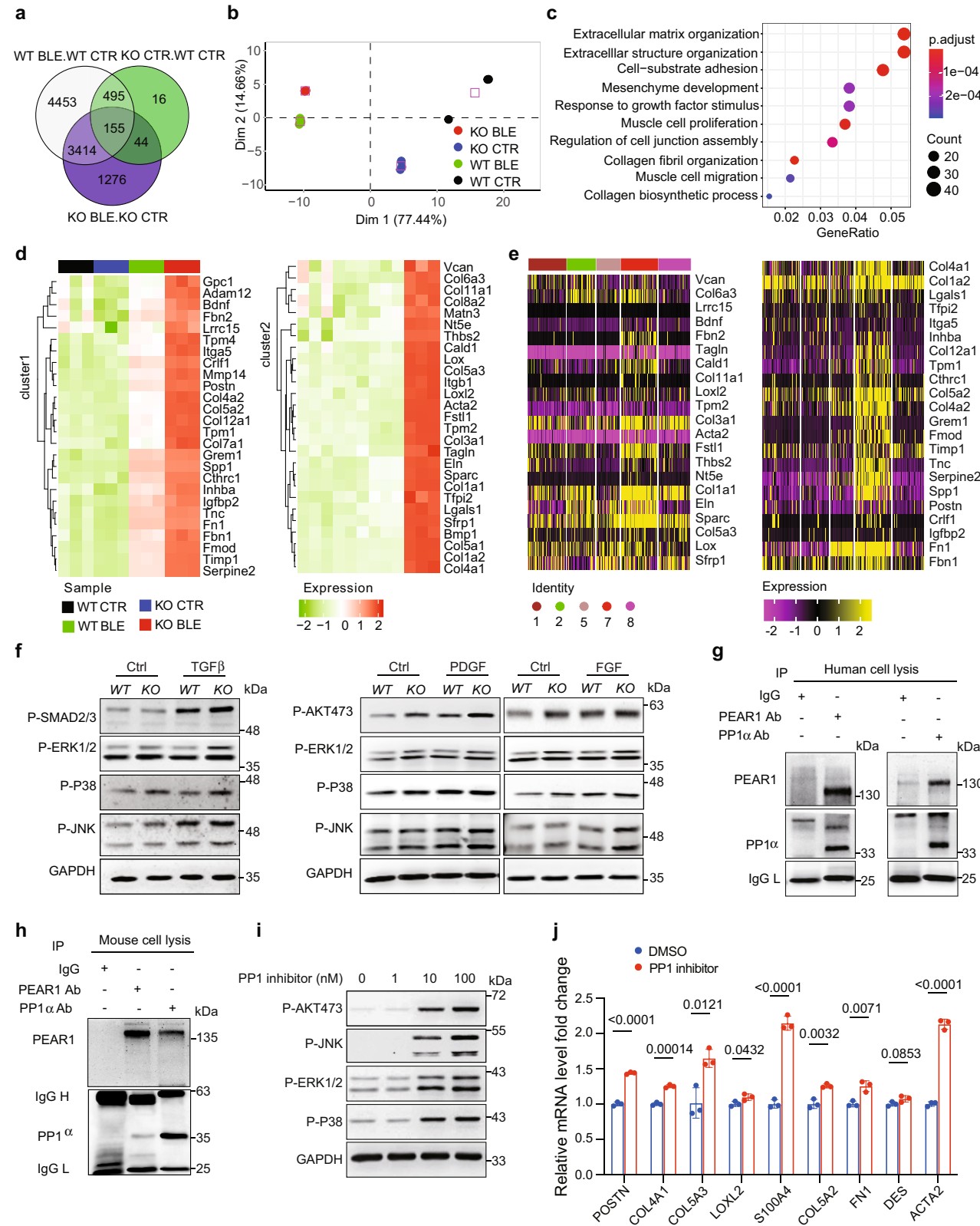

Exploration of the impact of PEAR1 on the origination and expansion of activated fibroblast is warranted in future study.

PEAR1 has been reported to associate with scaffold proteins Shc1 and Shc2 to regulate platelet activation[10]. In our study, we showed that PEAR1 might associate with PP1α, one of the three catalytic subunits of PP1, to regulate fibroblast function. PP1 is a major protein Ser/Thr phosphatase and is ubiquitously expressed in all eukaryotic

cells. Each functional PP1 enzyme consists of a catalytic subunit and a regulatory subunit[32]. These regulatory subunits, which contain a motif with the sequence of RVxF/W, may target the PP1 catalytic subunit to specific subcellular compartments, modulate substrate specificity, or serve as substrates themselves[33,34]. However, the intracellular sequence of human PEAR1 does not contain a regulatory motif, indicating that PEAR1 probably associates with a protein

**Fig. 3 | PEAR1 associated with PP1 to suppress fibrotic factor induced fibroblast activation.** Pdgfra⁺ cells were selected for bulk RNA-seq assay. **a** Overlapping pattern of the differentially expressed genes (DEGs) in three comparisons identified 155 core DEGs. **b** Principle component analysis (PCA) of replicated samples in each condition. Squares represent the centroid of corresponding sample group. **c** Top 10 GO biological process enriched with upregulated genes in *Pear1⁻/⁻* Bleo compared to *WT* Bleo. **d** Clustering of the leading-edge EMT genes in GSEA across 12 samples in four experiment conditions. Colored squares on bottom left represent experiment conditions. Colored bar on bottom right represents the scaled expression levels. **e** Leading-edge EMT genes in GSEA were predominantly expressed in cluster 7 (activated fibroblasts) cells of scRNA-seq data. Color bar on bottom left represents the scaled expression levels. Color dot on bottom left represents Pdgfra⁺ cell clusters in scRNA-seq data: 1-Col14a1 Matrix Fibroblasts, 2-Col14a1/Col13a1 Matrix Fibroblasts, 5-Col13a1 Matrix Fibroblasts, 7-Activated Fibroblasts, 8-Col14a1 Matrix Fibroblasts (Mmp3/Cyp1b1 high). **f** The phosphorylation levels of Smad2/3, AKT, ERK1/2, P38, and JNK1/2 were evaluated in *WT* and *Pear1⁻/⁻* fibroblasts stimulated by TGFβ, FGF, or PDGF for 30 min, respectively. GAPDH was used as a loading control. The experiments were repeated three times and the results were similar. **g, h** The binding of PEAR1 and PP1 were detected by immunoprecipitation in cultured human and mouse pulmonary fibroblasts. The experiments were repeated three times and the results were similar. **i** The phosphorylation levels of, AKT, P38, ERK1/2, and JNK1/2 were evaluated in fibroblasts incubated with 1 nM, 10 nM, or 100 nM PP1 inhibitor for 15 min, respectively. GAPDH was used as a loading control. The experiments were repeated three times and the results were similar. **j** The relative mRNA expression levels of ECM genes in cultured pulmonary fibroblasts incubated with PP1 inhibitor or DMSO as a negative control (*n* = 3 biologically independent samples in each group). For **c**, one-sided Fisher's exact test was used. For **j**, two-tailed t test was used. Data are presented as mean ± SD. Source data are provided as a Source data file.

containing a regulatory motif to mediate PP1 function. Further study is required to resolve this issue.

Inhalation therapeutics is not invasive and can deliver drugs directly to the lungs, limiting systemic exposure[35]. Here we show that the intratracheal nebulization of monoclonal antibodies targeting PEAR1 significantly ameliorates the degree of PF, indicating that the inhalation of antibodies may serve as an efficient strategy for the treatment of respiratory diseases.

## Methods

### Construction of mice

All animal experiments including euthanasia were in accordance with protocols approved by the Institutional Animal Care and Use Committee (IACUC) of Shanghai Jiao Tong University School of Medicine. All the mice were housed under a SPF condition (12-h light/dark cycle, 50% relative humidity, and 22 ± 2 °C) with free access to normal laboratory diet (SZS9126, Xietong Pharmaceutical Bioengineering, China) and water and monitored by inspection twice each day. All the mice used were C57BL/6 background unless otherwise indicated. For all experiments, 8–12-week-old male mice were used, unless otherwise indicated.

The *Pear1* knockout mice were constructed with the knockout-first strategy[36], generated by Cam-SU Genomic Resource Center (CAM-SU GRC). A cassette containing En2 SA, LacZ, Neo, FRT, and loxP sites, was inserted in introns between exon 6 and 7 of *Pear1* gene, and another loxP site are placed in intron between exon 8 and 9 of *Pear1* gene of (Supplementary Fig. 1a). The tm1a allele was initially as a *Pear1* non-expressive form which could trap the transcript through the En2 SA element and truncate it through the pA. *Pear1^{tm1c/+}* (*Pear1^{f/wt}*) allele was produced by crossing *Pear1^{tm1a/tm1a}* with a Flp transgenic mouse strain from Shanghai Biomodel Organism Science & Technology Development Co., Ltd (Shanghai, China).

Mesenchymal cell-specific *Pear1* deficient mice were generated by mating *Pear1^{tm1c/tm1c}* mice with a *Col1a2-cre^{ER}* transgenic mice[19] (from Professor Bin Zhou, Shanghai Institute of Biochemistry and Cell Biology) (Supplementary Fig. 2a). *Col1a2-cre^{ER}Pear1^{f/f}* mice (6 weeks old) were further injected intraperitoneally with tamoxifen daily (90 mg per kg body weight each time, 7 times) to generate mesenchymal cell-specific *Pear1* deficient mice.

Platelet-specific *Pear1* deficient mice were generated by mating *Pear1^{tm1c/tm1c}* mice with a *PF4-cre⁺* transgenic mice[37].

Human *Pear1* transgenic mice were constructed to integrate human *Pear1* cDNA to replace the mouse *Pear1* gene by CRISPR/cas9 technology by Bioray Laboratories. There are multiple transcripts of mouse *Pear1*, so the common exons 4–6 of mouse *Pear1* gene were selected to be replaced by human *Pear1* cDNA, and a SV40 poly A transcription termination signal after human *Pear1* cDNA was added to ensure that mouse *Pear1* is completely replaced by human *Pear1*. *Pear1*-L-sgRNA sequence is CCTCTGCAATGCCACTTTGTCCC. *Pear1*-R-sgRNA sequence is CCTGTGTCCGTGAGTCCTGAGTT. The sequences of primers for genotyping were provided in Supplementary Table 1.

### Bleomycin-induced mouse PF model

A mouse model of PF was established by intratracheal aerosolization of 1.5 μg/g (body weight) or 2 μg/g (body weight) of bleo (S1214, Selleck Chemicals). For antibody therapy, intratracheal aerosolization of antibody and IgG (401411, Biolgend) were administered on the 7th and 14th day after bleo induction. On the 21st day after bleo induction, the lung function and pathological changes of mice were analyzed.

### Amiodarone-induced mouse PF model

Mice were anaesthetized with isoflurane and then intubated oro-tracheally under a binocular loupe. Amiodarone (0.4 μg/g body weight or 0.8 μg/g body weight; A8423, Sigma) was administered every fifth day as an aerosol via microsprayer. For antibody therapy, intratracheal aerosolization of antibody were administered 7th and 14th day after amiodarone induction. On the 23st day after amiodarone induction, the lung function and pathological were analyzed. 8–12-week-old male mice were used.

### Measurement of lung function in mice

After anesthetized by intraperitoneal injection of pentobarbital (100 mg/kg), the mice were connected to a computer-controlled small animal ventilator through a tracheal tube according to the instructions of the manufacturer of anires2005 pulmonary function analysis system (version 2.0, bestlab, China). The change of pressure in the plethysmography chamber was measured by the port in the connecting tube with pressure sensor. FVC were used as indexes of pulmonary function.

### Isolation and culture of primary mouse lung fibroblasts

After the fresh lung tissue of mice treated with or without bleo was chopped, the tissue was digested in DMEM/F12 medium (Gibco) (containing 100 U/mL type I collagenase (C0130, Sigma), 2.5 mg/mL type IV collagenase (17104-019, Gibico), 0.3 mg/mL hyaluronidase (H4272, Sigma), and 0.1 mg/mL Dnase (DN25, Sigma)) for 2 h in 37 °C. 2% BSA was added and the suspension was centrifuged at 150 × *g* for 5 min. Then 10% fetal bovine serum DMEM/F12 medium was added, and cultured in a type I collagen (354236, Corning) coated cell culture dish at 37 °C and 5% CO₂ for 2 days. The primary lung fibroblasts were cultured no more than 3 passages.

### Flow cytometry

Fresh lung tissue was used for flow cytometry. The fresh lungs of mice were cut and digested to prepare single lung cells, which were collected after centrifugation and filtered by 40 μm cell strainer. The cell proportion was analyzed by flow cytometry using PerCP-Cy™5.5 rat anti-mouse CD45 (561869, BD Biosciences, 1:100), PE/Cy7 anti-mouse CD31 Antibody (102418, Biolegend, 1:100), BV421 rat anti-mouse

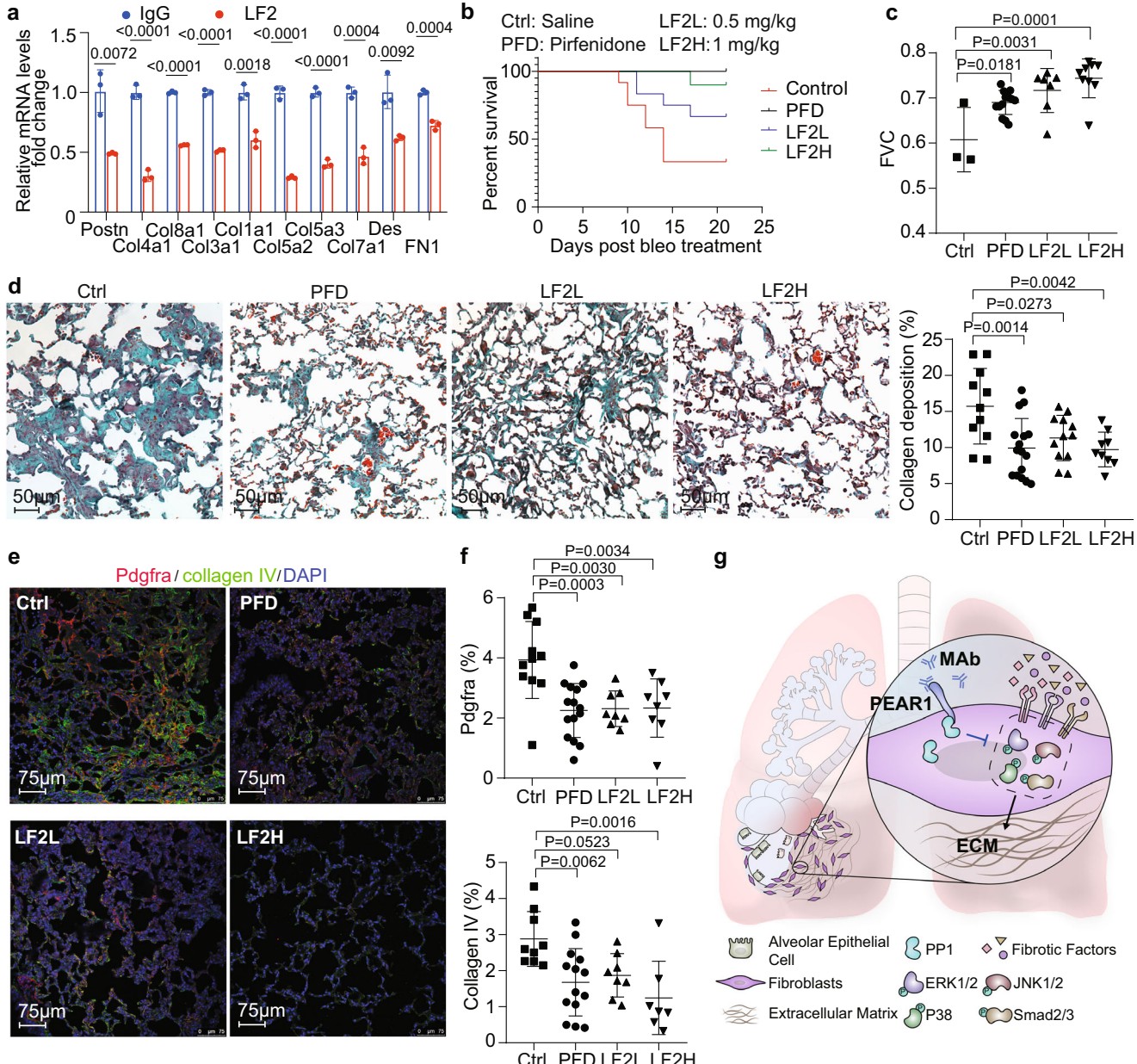

**Fig. 4 | Monoclonal antibody targeted human PEAR1 for PF therapy. a** The mRNA expression levels of extracellular matrix genes in human pulmonary fibroblasts (HFL1) treated with 1 μg/mL anti-human PEAR1 monoclonal antibodies (LF2) by qPCR (n = 3 biologically independent samples in each group). LF2 is a humanized antibody with an ADCC/ADCP-weak human IgG4 Fc with S228P mutation. **b** Survival curves of humanized *Pear1* mice induced by 1.5 μg/g (body weight) bleo treated with 300 mg/kg pirfenidone daily by gavage (PFD), 0.5 mg/kg (LF2L), 1 mg/kg (LF2H) LF2 through the trachea once a week. Saline was used as control (Ctrl) (n = 11 mice in Ctrl group; n = 16 mice in PFD group; n = 12 mice in LF2L group and n = 10 mice in LF2H group). **c** FVC was measured of Ctrl group mice, PFD group mice, LF2L group mice, and LF2H group mice (n = 3 mice in Ctrl group; n = 11 mice in PFD group; n = 7 mice in LF2L group; n = 9 mice in LF2H group). **d** Representative images of masson staining on lung sections from Ctrl group mice, PFD group mice,

LF2L group mice, and LF2H group mice. The collagen area (green) was calculated and statistics were performed (n = 11 mice in Ctrl group; n = 16 mice in PFD group; n = 12 mice in LF2L group; n = 9 mice in LF2H group) (scale bars, 50 μm). **e**, **f** Representative images of immunofluorescence staining on lung sections from Ctrl group mice, PFD group mice, LF2L group mice, and LF2H group mice for Pdgfra (red), collagen IV (green), and DAPI (blue). The fluorescence positive area was calculated and statistics were performed (n = 11 mice in saline control group; n = 15 mice in PFD group; n = 8 mice in LF2L group; n = 8 mice in LF2H group) (scale bars, 75 μm). **g** A schematic diagram of the mechanism of PEAR1 in regulation of pulmonary fibrosis. ECM, extracellular matrix; MAb, monoclonal antibody. For **c**, **d**, **f**, one-way ANOVA was used. Data are presented as mean ± SD. Source data are provided as a Source data file.

---

CD326 (563214, BD Biosciences, 1:100), PE anti-mouse Pdgfra (12-1401-81, Thermo Fisher, 1:100) or APC anti-mouse Pear1 (labeled APC in our lab with the labeling kit from Thermo Fisher #A20186). The expression of human PEAR1 on humanized mouse primary fibroblast was measured by flow cytometry using APC anti-human PEAR1 (FAB4527R, R&D

Systems, 1:100). All the flow cytometry experiments were carried out on aria III, a public technology platform of Shanghai Jiaotong University. The flow cytometry data were analyzed using FlowJo software. The gating strategy of flow cytometry was provided in Supplementary Fig. 14.

## Masson staining

The lung samples were stained with the hematoxylin (517-28-2, Sangon, China) for 5 min. Then the sections were differentiated with ethanolic hydrochloric acid and rinsed to blue-black with flowing water; the Ponceau 2R (A600753, Sangon, China) and fuchsin acid (A610469, Sangon, China) was used to stain for 10 min, then sections were rinsed with distilled water, and differentiated with 1% phosphomolybdic acid (A600709, Sangon, China) for 2 min. The sections were stained with the brilliant green (A6110007, Sangon, China) for 5 min and were sealed with optical rubber. The sections were observed under a light microscope, and the image data were collected to calculate the volume fraction of collagen (The area of collagen deposition (%) = average collagen area/area of total field × 100).

## Plasmid construction and protein expression

The expression plasmids of 6×His tags human and mouse PEAR1 ECD (Extracellular domains) were constructed in pcdna3.1. Then the HEK293S cells were transfected by the plasmid with PEI (24765, Polyscience, China) and cultured in a serum-free suspension. The supernatant was extracted and the protein was purified by nickel column.

## Immunoprecipitation

Human or mouse fibroblasts were incubated in Cell lysis buffer (P0013J, Beyotime, China) with protease inhibitor (14001, Bimake) at 4 °C for 15 min and then centrifuged at $12,000 \times g$ for 15 min. The lysates were then incubated with anti-human PEAR1 monoclonal antibody LF2 and LF3, anti-mouse PEAR1 monoclonal antibody LF1, anti-PP1 antibody (SC7482, Santa Cruze) overnight at 4 °C, respectively. Mouse IgG (5415, Cell Signaling Technology) was used as a negative control. On the next day, protein A/G agarose beads (SC2003, Santa Cruze) were added to the lysates and incubated for 3 h. Then the agarose beads were harvested and washed 3 times with PBS by centrifugation at $12,000 \times g$ for 1 min, followed by boiling for 5 min at 100 °C in Samples were analyzed by Western blotting with the indicated antibodies.

## Detection of relevant protein expression by western blot

The primary fibroblasts were treated with TGFβ, FGF, PDGF (7666-MB, 3139-FB, 1447-PC, R&D Systems), or PP1 inhibitor Calyculin A (141784, Abcam) for 15 or 30 min, respectively. After 2 times of PBS washing, the total protein was extracted by lysate with protease inhibitor and phosphatase inhibitor. The total protein concentration was detected by BCA kit, and the protein was separated by 10% SDS-PAGE and then transferred to PVDF membrane. The 5% skimmed milk was used to block the protein at room temperature for 2 h, and then the primary antibodies were added and reacted at 4 °C overnight. On the second day, the corresponding second antibody was added and sealed at room temperature for 1 h, followed by the final step of ECL addition for exposure. The antibodies contain anti-p-Smad2/3 antibody (8828, Cell Signaling Technology, 1:1 K), anti-p-P38 antibody (9211, Cell Signaling Technology, 1:1 K), anti-p-ERK1/2 antibody (4370, Cell Signaling Technology, 1:2 K), anti-p-AKT antibody (4060, Cell Signaling Technology, 1:2 K), p-JNK antibody (4668, Cell Signaling Technology, 1:1 K), anti-GAPDH antibody (30201ES, Yeasen, China, 1:1 W), anti-rabbit IgG secondary antibody (305-035-003, Jackson ImmunoResearch, 1:1 W) and anti-mouse IgG secondary antibody (115-035-003, Jackson ImmunoResearch, 1:5 K).

## Platelet function assay

For platelet aggregation experiment, the washed platelets from *WT* and *Pear1*$^{-/-}$ mice were diluted to $3 \times 10^8$/mL with tyrode's buffer. And the 300 μL platelet concentrates were stimulated with 1.5 μg/mL collagen (P/N385, Chrono-log) or 0.05 U/mL thrombin (HT4082A, Enzyme Research Laboratories) and measured using an aggregometer (Chrono-Log) according to the manufacturer's protocol.

For detection of P-selectin exposure and JON/A binding in platelet, platelets ($5 \times 10^7$ platelets/mL) were stained with FITC-conjugated anti-mouse P-selectin antibody (553744, BD Biosciences, 1:100) or JON/A(PE) (M023-2, Emfret Analytics, 1:50) for 20 min at 37 °C in a 50 μL volume. The reaction was stopped by addition of 450 μL tyrode's buffer and then analyzed by a flow cytometer.

For clot retraction, mouse platelets were processed as previously described[38]. Clot size was quantified from photographs using National Institutes of Health Image J software, and retraction was expressed as retraction ratio (1-[final clot size/initial clot size]).

## Matrigel tube formation

The capacity of pulmonary endothelial cells from *WT* and *Pear1*$^{-/-}$ mice to form capillary-like structures (tubes) was evaluated using 96-well plates coated with BD Matrigel™ Basement Membrane Matrix (BD Biosciences). Endothelial cells were digested with accutase (Yeasen, China) and were seeded in Matrigel-coated wells at a density of $1 \times 10^4$ cells/well and maintained in 1% FBS growth medium. Tubes were visualized using a Zeiss microscope and photographed. The total tube length was measured using Image J software.

## Single-cell RNA sequencing (scRNA-seq) assay

Fresh lung tissues derived from 5 mice per group were pooled for tissue digestion and single-cell isolation. All of the single-cell samples were processed in the same batch for tissue collection, flow cytometry sorting, single-cell isolation and library preparation, and sequencing. Mesenchymal cells (CD31⁻CD45⁻CD326⁻) were isolated from fresh lung using a negative sorting approach as described previously[21,39]. scRNA-seq assay was performed in Novogene (Nanjing, China). Single-cell suspensions were processed using the 10x Genomics Single Cell 3' v3 RNA-seq kit. Cellular suspensions were loaded on a Chromium Controller instrument (10x Genomics) to generate single-cell Gel-Bead-in-Emulsions (GEMs), followed by single-cell partitioning and barcoding. RNA from the barcoded cells was subsequently reverse-transcribed and indexed for library construction according to the manufacturer's instructions. Final libraries were sequenced on the NovaSeq platform (Illumina) to reach 98k–117k reads/cell. In summary, a total of 3.5G base paired-end reads were generated to screen 33355 cells.

## scRNA-seq raw data processing and unsupervised cell classification

We used Cell Ranger version 3.1 (10x Genomics) with STAR (version 2.7.2b) to process raw sequencing data in fastq format and align to mouse genome assembly GRCm38 (mm10). Seurat suite version 3.0 was used for downstream quality check, cell and gene feature filtering, and unsupervised clustering of cells[40]. Cells with >10% mitochondrial genes were considered as dead or stressed-out cells, and thus were removed. There was minimal ambient RNA contamination. We removed under-sequenced cells with nFeature_RNA < 300. To restrict doublets, we further removed cells with nFeature_RNA > 3500–5000 according to diverse sequencing depth in each sample. The 24,640 cells surviving quality control were combined, normalized, and classified into 18 clusters for cell marker phenotyping (Supplementary Fig. 3). We found 29.4% (7246/24640) of total cells were positive with at least one of CD31 (Pecam1), CD45 (Ptprc), or CD326 (Epcam) (Supplementary Fig. 3a–c, respectively). These 7246 cells in Cluster 1, 3, 5, 7–10, 13–17 were considered as leakage during flow cytometry sorting, and thus were removed from downstream analysis (Supplementary Fig. 3a–c).

After quality control and cell filtering, 17,934 mesenchymal cells were used for downstream data analysis. Cells across all experimental groups were combined and normalized to ensure the same sequencing depth. We used the first 20 principal components to perform UMAP (Uniform Manifold Approximation and Projection) and tSNE (t-distributed stochastic neighbor) analysis for dimension reduction and

visualization. Seurat function 'FindClusters' with parameters "resolution = 0.8" and "dims.use = 1:20" were adopted in unsupervised clustering.

## Supervised annotation of single-cell clusters

We used three consecutive levels of supervised approach for single-cell annotation. We firstly used R package "scMCA" to define cell types in mouse based on single-cell digital expression in Mouse Cell Atlas v2.0 (http://bis.zju.edu.cn/MCA/)[40]. Because MCA v2.0 is derived from normal mouse cells, we next incorporated mouse lung mesenchymal cell types involved in bleo model of PF in our cell annotation. "CellAssign" is a probabilistic model that leverages prior knowledge of cell-type marker genes to annotate single-cell RNA sequencing data into predefined or de novo cell types[41]. It assigns cell types through unsupervised clustering followed by manual annotation or via 'mapping' to existing data. We retrieved cell type markers from previously published mouse bloe models[20,21] and compiled into reference dataset for "CellAssign" mapping with criterion of probability ≥90%. Finally, due to the heterogeneous progenitors and sources of myofibroblasts, we used the molecular cell atlas of the human lung to manually refine the Acta2+ (α-SMA+) cell types defined by MCA v2.0[23]. Marker genes across all clusters or between two clusters were identified using Seurat 'FindMarkers' with parameter "min.pct=0.5".

## Bulk tissue RNA-Seq

We used flow cytometry to sort out Pdgfra+ cells from mouse lungs. Total RNAs were isolated from the sorted cells with TRIzol™ reagent (Invitrogen) according to the manufacturer's protocol. RNA quality was evaluated using Bioanalyzer (Agilent, Santa Clara, CA). All RNA samples displayed RNA Integrity Number (RIN) > 8.5. SMART-Seq v4 Ultra Low Input RNA Kit (Clontech Laboratories, Inc.) was used for generating high-quality, full-length cDNA from low amount of RNA. TruePrep TM DNA Library Prep Kit V2 for Illumina (Vazyme TD502) was used for library construction. HiSeq X10 from Illumina was used for sequencing, reads mode is double-ended queue 151 bp, quantity is Q30 > = 90%.

## RNA-seq raw data process

Raw reads in .fastq format were aligned to the mouse genome assembly GRCm38 (mm10) using HISAT2 RNA-sequencing alignment software[42]. The aligned .bam files were mapped to mouse reference transcriptome (version mm10). The abundance of transcripts was summarized into CPM (counts-per-million reads). Raw counts data were normalized using variance stabilizing transformation (VST) method implemented in R/Bioconductor "DEseq2" package[43].

## Sample classification and differential expression profiling

Sample and gene clustering were performed using R/Bioconductor package 'ComplexHeatmap', and "Seurat" function "DoHeatmap". Differentially expressed genes (DEGs) between experiment groups were identified using the empirical Bayes (eBayes) moderated two-sided t-test implemented in the R/Bioconductor package "limma"[44]. The criteria for significance were set to fold change >2 and false discovery rate (FDR) < 0.05 for multiple comparisons adjustment. DEGs between two experiment conditions were listed in Supplementary Data 1.

## Pathway analysis

We used R package "rWikiPathways" for pathway analysis of cluster markers or DEGs between experiment conditions[45]. Top 10 Biological Process of Gene Ontology (GO) and KEGG pathways enriched with DEGs were reported for the selected clusters or condition with criterion of with FDR < 0.01. Threshold-free pathway analysis based on the differential expression rank of all genes was performed using R package "rWikiPathways" or Gene Set Enrichment Analysis (GSEA) software[46].

## The preparation of monoclonal antibody for mouse and human PEAR1

We first constructed a plasmid, transfected the cultured HEK293S cells to express the 6×his labeled mouse PEAR1 extracellular segment (leu19-ser754) protein, purified by nickel column, and then immunized Pear1−/− mice with immune adjuvant. After the standard immune process, the spleen of mice was isolated, and then the monoclonal hybridoma was prepared by standard single clone antibody preparation technology, and enzyme-linked immunosorbent assay (ELISA) was used to screen monoclonal antibody hybridoma which can bind to mouse PEAR1. There were 137 monoclonal hybridoma cell lines which can secrete and bind to mouse PEAR1. By using qPCR method to detect the effect of antibody on the expression level of extracellular matrix protein in mouse pulmonary fibroblasts in vitro, it was found that LF1, a monoclonal antibody, could significantly suppress the synthesis of multiple ECMs by pulmonary fibroblasts. We also constructed a 6×his labeled human PEAR1 extracellular segment (leu21-ser754) plasmid, transfected HEK293S cells in suspension culture, purified by nickel column, mixed with immune adjuvant, immunized BALB/c mice, separated the spleen of mice through immune standard process, and then prepared monoclonal hybridoma cells by standard single clone antibody preparation technology, and screened the hybridoma cells that can combine with human PEAR1 by ELISA. Among them, 53 monoclonal hybridoma cell lines can secrete antibodies binding to human PEAR1. Among them, we found that 14 antibodies can suppress the expression of extracellular matrix protein in human lung fibroblasts. Then, in order to reduce the immunogenicity of the antibody in human, one antibody named LF2 was humanized. In order to avoid the platelet clearance induced by antibody-mediated ADCC/ADCP effect and resulting in serious side effects, the human IgG4 subtype was selected. And the S228P mutation was introduced into the Fc region to prevent the fab change and stabilize the IgG4 molecule[47].

## Evaluation of the function of LF2 in human fibroblast in vitro

Normal lung fibroblasts (HFL1; ATCC, CCL-153) and PF fibroblasts HFL1 (ATCC; PCS-201-020) were used to detect the function of LF2 mab in vitro. PF fibroblasts and HFL1 were serum-starved for 12–16 h when the cells were grown to 60% confluence. Then cells were pre-incubated for 1 h with LF2 before the stimuli of human TGFβ (10 ng/ml, 240-B, R&D Systems). After treatment for 48 h, the relative expression level of mRNA (POSTN, TIMP1, α-SMA, FN1) was detected. This study was approved by the Institutional Review Board on Human Subjects Research and Ethics Committees (Shanghai Jiao Tong University School of Medicine).

## Statistical method

For scRNA sequencing statistical analysis, Pathway over-representation analysis was performed using R package "rWikiPathways"[45]. P values are calculated using one-sided Fisher's exact test. Multiple comparisons are adjusted using Benjamini−Hochberg method. DEGs between experiment groups were identified using the empirical Bayes (eBayes) moderated two-sided t-test implemented in the R/Bioconductor package "limma"[44]. For statistical analyses of other experiment results, Prism 8 software was used. Two-tailed unpaired Student's t test was used for the comparison between two groups. Comparisons among multiple groups were performed using one-way ANOVA followed by Tukey multiple comparison test. Data are shown as mean ± standard deviation (SD). In all data comparison, P value less than 0.05 is considered statistically significant.

## Reporting summary

Further information on research design is available in the Nature Portfolio Reporting Summary linked to this article.

## Data availability

The raw reads for scRNA-seq in fastq format have been deposited in Sequence Read Archive (SRA) database under accession number SRP335961. The processed data and meta data that support the findings of this study have been deposited in the Gene Expression Omnibus (GEO) under accession number GSE183545. The raw reads data for bulk tissue RNA-seq were deposited in Sequence Read Archive (SRA) database under BioProject accession number PRJNA749378, and the raw expression data and normalized data were deposited in Gene Expression Omnibus (GEO) under accession number GSE183657. All other data are available in the article and its supplementary files or from the corresponding author upon request. Source data are provided with this paper.

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

## Acknowledgements

The work was funded in part by the National Key R&D Program of China (2021YFA0804900 to J.L.), National Natural Science Foundation of China (grants 82030004, 31830050, 81721004 to J.L.; grants 81970123 to X.F.), Innovative research team of high-level local universities in Shanghai (SHSMU-ZDCX20211801), and National Institute of Health (grants RO1HL130796 and UG3HL145266 to I.N.). We also thank Yun Zhou from Shanghai Jiao Tong University School of Medicine for helping develop the monoclonal antibody. The authors thank members of Cambridge-SU Genomic Resource Center for generating knockout mice supported by grant from National Key R&D Program of China (2018YFA0801100). The authors also thank Professor Bin Zhou from Shanghai Institute of Biochemistry and Cell Biology for the gift of *Col1a2-cre^{ER}* transgenic mice.

## Author contributions

X.F. and J.L. conceptualized and supervised the study. Y.H. and Y.G. conducted most of the bioinformatics analysis. Y.G. and H.G. performed expression and purification of PEAR1 protein. X.F., K.L., L.L., and Y.G. developed the monoclonal antibodies. X.F., Y.G., and L.L. performed the experiments and analyzed data. J.Y., A.Y., L.Z., Y.S., and X.W. helped with the experiments. J.L., X.F., and Y.H. wrote the paper with input from all authors. I.N. helped revise the manuscript.

## Competing interests

The authors have submitted a patent application (application numbers PCT/CN2021/073547, CN202011212639.8) based on the results reported in this study. And the authors declare no other competing interests.
