## [Peer Review File · Nature Communications]

PEAR1 Regulates Expansion of Activated Fibroblasts and Deposition of Extracellular Matrix in Pulmonary FibrosisREVIEWER COMMENTS

Reviewer #1 (Remarks to the Author):

This manuscript describes the role of Pear1 in activated fibroblasts involved in fibrosis. The authors provide both in vitro and in vivo support for their hypothesis. The findings are interesting, but several gaps/points need to be addressed to support the authors claims.

1) The data for Pear1 IP experiments (Fig. 3g/h) do not agree with the text. In Fig. 3g similar amounts of PEAR1 are detected with IP of PEAR1 or PP1a, implying that all of PP1a is bound to PEAR1. However, with the PEAR1 IP no PP1a is detected. Please clarify.

2) For the in vivo study, please provide some rationale for doses selected. 0.5 vs. 1 mg/kg seems like an arbitrary distinction yet there seems to be significant differences (Fig. 4b, c, d). Please define AbL and AbH, LF2L, LF2H. Provide additional information on isotope control (isotope, species). Please define the isotope for LF1 and LF2.

3) Since Pear1 is a membrane protein, the effects seem in the mouse study could be due to depletion of Pear1+ cells via ADCC/ADCP or the direct functional effect as suggested by the authors. To support the claim that Pear1 agonism is the relevant MOA, the authors should show data with an effectorless Fc or some other additional data.

4) Manuscript needs grammar editing.

Reviewer #2 (Remarks to the Author):

While platelet function does not appear to be affected (Supp Fig 10) since PEAR1 signalling is activated upon platelet aggregation and independent of activation, this suggests that platelets may be directly involved in the progression of Bleo pulmonary Fibrosis and this aspect of platelet biology was earlier reported by Piguet in 1994. Authors should consider these earlier publications and incorporate their current findings as to the direct involvement of non-activated but aggregated platelets in the progression or resolution of fibrosis.

1 Piguet PF and Vesin C. Pulmonary platelet trapping induced by bleomycin: correlation with fibrosis and involvement of the beta 2 integrins. *Int J Exp Pathol* 1994;75(5):321-8.

2 Piguet PF, Tacchini-Cottier F, Vesin C. Administration of anti-TNF-alpha or anti-CD11a antibodies to normal adult mice decreases lung and bone collagen content: evidence for an effect on platelet consumption. *Am J Respir Cell Mol Biol*. 1995;12(2):227-31.

3 Piguet PF, Van G, Guo J. Heparin attenuates bleomycin but not silica-induced pulmonary fibrosis in mice: possible relationship with involvement of myofibroblasts in bleomycin, and fibroblasts in silica-induced fibrosis. *Int J Exp Pathol* 1996;77(4):155-161.

There does not appear to be any data on the effect of the mAb raised against the humanized PEAR1 on human fibroblasts that are readily accessible. Moreover, one may expect to see differences in response when fibroblasts from established fibrotic lungs are compared to normal "control" human lung fibroblasts. Since most patients presenting with PF have established fibrotic tissue, whether the cells in the fibrotic lung would respond to the antibody exposure would be an important issue.

Do the authors have any data regarding the penetration by the LF1 antibody of epithelial cells or cell junctions to enter the interstitial tissue to access the fibroblasts and myofibroblasts present there?? Is the antibody acting at the epithelial cell to inhibit EMT? Does the antibody inhibit myofibroblast differentiation from resident fibroblasts (after stimulation with factors such as TGFb)?

What does PEAR1 deficiency in aging mice do to staining for aSMA and detection of myofibroblasts?

In relation to the early work of Piguet and colleagues, is this involvement of PEAR1 exclusive to bleomycin treatment (especially relevant to ref 3 by Piguet) ?? Many drugs showing effects in bleomycin induced experimental fibrosis have failed in moving into human studies.

Reviewer #3 (Remarks to the Author):

This is an interesting study on fibroblast biology. The authors report that Pear1 in FBs is a new molecular target for PF therapy. The authors showed that Pear1 deficiency caused respiratory function decline in aged mice, and higher degree of PF and mortality in bleo model. Single cell RNA-seq analysis of pulmonary FB revealed a potential function of Activated-FB cluster. Intratracheal aerosolization of Pear1 monoclonal antibody ameliorates PF in both wild-type and Pear1-humanized mice. Fibroblast are not easy to study. But the authors did a great job investigating into a challenging field. I only have a few comments:

1. The authors may improve the presentation of their manuscript by using NPG Language Editing service.

For example:

"IPF has a poor prognosis, is unpredictable, leads to irreversible progression and eventually respiratory failure and death"

"It was showed that Pear1 is also expressed in endothelial cells"

"Moreover, isolated and cultured FBs from WT lung tissue were almost all expression of Pear1"

"Observation of aged mice, we found that....."

2. It is confusing why monoclonal antibody targeting human Pear1 actually activated pear1. More explanation is needed.

3. Biological replicates and batch information in the single cell mRNA-seq experiment should be presented.

4. Sftpc and Cyp2e1 are markers of lung epithelial cells. However, the authors labelled cluster 3 and cluster 0 as stromal cells? Any contamination?

5. Based on Figure 2a, Cluster 6 includes two separate groups. Why are they defined as one cluster?

6. Figure 3g: PP1a Ab IP generates more Pear1 signal than Pear1 Ab IP. Why?

7. How specific is Pear1 antibody? The left side of hPEAR1 signal in Extended Data Fig.9b looks a bit different from the other areas. In addition, the voltage is quite high for PEAR1 signal in Extended Data Fig.9c. Is it because of a problem with the antibody?

Reviewer #4 (Remarks to the Author):

This challenging study describes a role for PEAR1 in epithelial mesenchymal cell transition into lung fibroblasts, in *Pear1*^{-/-} mouse lungs, leading to progression of pulmonary fibroblasts, expressing a series of lung fibroblast markers. The well-designed and technically solid study shows that the phenotype observed in *Pear1* deficient mice can be recapitulated in a cell-specific mouse K.O., which excludes the option that the transformations observed in aging *Pear1*^{-/-} mice and in bleomycin-treated *Pear1*^{-/-} mice, would be caused by a variety of lung cells indirectly. This study, importantly, shows that PEAR1 plays a role in other cells than the few having been identified, so far.

Detailed remarks:

1. Line 24 and next: In the introduction the authors state that *Pear1* may activate platelets by binding its EMI domain to an unknown ligand. This is not entirely correct: a putative ligand has been identified, the high affinity immunoglobulin E receptor subunit alpha (*FcεR1α*), present on human platelets, and involved in human platelet activation amplification via *PEAR1*-*FcεR1α* interactions. On murine platelets, both *Pear1* and especially *FcεR1α* are expressed in lower amounts than on human platelets, strongly reducing contributions by *Pear1*-*FcεR1α* interactions to platelet activation, to insignificant levels. So, the statement in line 24 about "*Pear1*" would be "partly" correct for human PEAR1. Murine *Pear1* deficiency does not affect platelet function, induced by regular platelet agonists, but semi-specific *Pear1* agonists, such as dextran sulfate and fucoidan can potently activate mouse platelets, in a *Pear1*-dependent manner. It is surely correct that *FcεR1α* may not be the primary or major receptor and other ligands or receptors may be identified in the future. So, these phrases might be slightly adapted (murine *Pear1* vs. human PEAR1).

2. The present study specifies mesenchymal cell proliferation but it is not clear whether this study describes a proliferation or rather a differentiation study, or the mixture of both. In megakaryocyte progenitors and endothelial cells, PEAR1 gene-deficiency enhances cellular proliferation in vitro and in vivo, which has been coupled to increased Akt-phosphorylation and reduced PTEN expression. In those studies, the reduced expression of PTEN in *Pear1*^{-/-} mice has been advocated to explain the proliferative phenotype. As is also found in the present work, *Pear1* does not play a role in developmental angiogenesis, but in models of neoangiogenesis and wound healing, *Pear1* plays a clear role. Did the authors in the present work investigate cellular proliferation in their lung sections (use of cellular proliferation markers, such as PCNA or Ki67) and did they look into Akt-P levels during mouse aging and/or during bleomycin treatment? Did they do co-staining with the epithelial, mesenchymal or fibroblast markers? That will determine whether mesenchymal cell proliferation is the driver or not. In the extended part, several lung sections are stained for specific markers. It would be very informative to combine some of this staining with examples where a proliferation marker would be used, such as PCNA or Ki67 to determine whether the cellular differentiation, presently reported coincides or not with increased proliferation of mesenchymal cells, or fibroblasts. That will clarify whether the increased deposition of ECM components results from a transdifferentiating effect or a combined proliferative and/or functional effect.

3. Interestingly, in this study, the authors found an association between *Pear1* and Protein Phosphatase 1, but did they do (semi)-quantitative analysis of its presence in lung sections of WT and *Pear1*^{-/-} mice? Or staining/western blots? Does its concentration shift during aging or bleomycin treatment? The authors now invoke phosphorylation of several growth factors to explain the cellular differentiation process, but can they exclude that differential Protein Phosphatase 1 in WT and *Pear1*^{-/-} mouse lungs are responsible for differential states of baseline Akt phosphorylation in both types of lung, driving cellular reorientation differently, as a result of different metabolism of the differentiated fibroblasts? PTEN has been reported to play a crucial role in endothelial cells in angiogenesis, but it is not impossible that in mesenchymal cells/fibroblasts, a similar role exists for the PP1α subunit, especially since PTEN is not picked up in the RNA sequencing. Since Protein Phosphatase associates with *Pear1*, what is its fate in the *Pear1* K.O. lung cells?

4. In platelets and endothelial cells, PEAR1 activation with an antibody triggers rapid PEAR1 phosphorylation and activation of Akt and all of its coupled pathways. (These will differ from cell to

cell.) Yet, this action is a cell signaling response to an external activation signal, meaning that this is transient. In contrast, in PEAR1 gene-deficient cells, there is a permanent impact of the gene-deficiency on several pathways and on gene-regulation and protein expression. It is difficult to understand how the instillation of an activating anti-Pear1 antibody can control the "same" cellular differentiation cascade as found during gene-deficiency. The antibody is given with an interval of one week. How long does the effect of the antibody last, as measured e.g. by PEAR1 or Akt phosphorylation or phosphorylation of any other signaling molecules, or Protein Phosphatase 1? The results are clear: there is a correction of the phenotype, but is this the result of a signaling cascade triggered by the antibody that is controlling the same pathways as those that are dysregulated during bleomycin treatment? Are these the same pathways, as those that are affected in the K.O. cells?

5. The aging phenotype is investigated in 12 months old mice. How old were the mice subjected to bleomycin treatment? Same question for the Col1a2-CreERPear1f/f mice? Is the age the same in all experiments? Is there a gender dependence, or is the same sex used throughout?

6. Already in the abstract, the authors state "Mesenchyme-specific Pear1 deficiency aggravated bleomycin-induced PF, confirming that Pear1 modulates PF progression potentially via regulation of FB function". This phrase leads to some confusion, because the notion of mesenchymal cell function and fibroblast function are mixed. The authors should provide their definition of a mesenchymal cell in the Extended data part, supported (and explained) by the corresponding flow cytometric definition. Flow cytometric definition is given in the legend of Figure 1: epithelial cells are defined as "CD326+CD45-CD31-" and fibroblasts as "Pdgfra+CD45-CD31-". Mesenchymal cells are defined as "Pdgfra+CD31-CD45-CD326-". However, in the Results page 4, line 4, mesenchymal cells are defined as "Pdgfra+CD31-CD45-", i.e. irrespective of their CD326 presence. In some places, mesenchymal cells are defined as Pdgfra+ only. Some clarification about the relation between lung mesenchymal cells, epithelial cells and fibroblasts, as detected in the present study by flow cytometry will avoid confusion and will make clear whether (activated) fibroblasts are meant or mesenchymal cells. Along these lines it would be informative to show in Fig 1f the difference for WT CD31-CD45- and Pear1-/- CD31-CD45- at the representative age used (see question 5) and 12 months of age. Encircle the relevant cell populations and report their proportions and distribution differences, using the standardized definitions.

7. As mentioned above, it is important to distinguish between a model in which the increased PF is caused by enhanced proliferation of mesenchymal cells, subsequently translating in more fibroblasts and more deposition of EC components and a model in which activity dysregulation in mesenchymal cells will translate in more transformation into "activated" fibroblasts. The authors show that Protein Phosphatase associates with Pear1. How sure is it that this phosphatase suppresses fibrotic factors-induced intracellular signaling, as this is shown for TGFβ, PDGF and FGF? Fig 3f shows no differences between WT and KO bands after stimulation. Also, there appears to be little difference between WT and K.O. controls. Yet, Akt-P in Ctrl KO and after FGF appear to be comparable and higher than in the WT condition. Is this correct? In that case, this analysis would be parallel to what is seen in endothelial cells, where the knock-down or knock-out raises the baseline phosphorylation of Akt, thus generating a proliferative phenotype. It would be interesting to have some measurement of Protein Phosphatase 1 in WT vs KO fibroblasts, maybe with a similar analysis for PTEN (maybe not when the authors can confirm from their RNA sequencing data that it does not play a role in the present report). In that case, the enhanced Akt-phosphorylation reflects a knock-out related genetic and biochemical change, in turn driving all the aspects related to the changed phenotype.

8. The PP1 inhibitor has a very pronounced effect on the phosphorylated bands, which reflects the importance of this phosphatase. The evident question here is: does this inhibitor (in the adequate concentration-range) have the same effect for WT and KO cells, before and after bleomycin treatment? Moreover, does it have an effect on the baseline phosphorylation of PEAR1 itself?

9. The English language requires some attention and should be checked by a native English speaker.

Point by point response to the reviewers' comments:

We would like to take this opportunity to thank the reviewers for their in-depth constructive comments on our work. We believe your suggestions helped to greatly strengthen the manuscript.

To Reviewer #1

1. The data for Pear1 IP experiments (Fig. 3g/h) do not agree with the text. In Fig. 3g similar amounts of PEAR1 are detected with IP of PEAR1 or PP1 α , implying that all of PP1 α is bound to PEAR1. However, with the PEAR1 IP no PP1 α is detected. Please clarify.

Response: Thank you for your careful review. As you pointed out, when using PEAR1 Ab IP, the PP1 α band was weak. Since the band of PEAR1 which was pulled down by this antibody was also weak, we speculate that this may be due to the low enrichment efficiency of the PEAR1 antibody we originally used. Therefore, several kinds of other antibodies have been tried. It can be seen from the new Fig. 3g the band of PEAR1 which was pulled down by the new antibody was much deeper, and the PP1 α band immunoprecipitated was obvious, correspondingly.

2) For the *in vivo* study, please provide some rationale for doses selected. 0.5 vs. 1 mg/kg seems like an arbitrary distinction yet there seems to be significant differences (Fig. 4b, c, d). Please define AbL and AbH, LF2L, LF2H. Provide additional information on isotope control (isotope, species). Please define the isotope for LF1 and LF2.

Response: Thanks for your professional suggestions. How to deduce *in vivo* pharmacodynamic doses from *in vitro* cell function experimental doses is a difficult question to answer. Therefore, in the early stage, we used a small number of animals to do a groping experiment on the appropriate dose *in vivo*, and the dose was set to 1 mg/kg and 5 mg/kg. Both doses were effective in improving pulmonary fibrosis, and there was no significant difference in the efficacy between 1 mg/kg and 5 mg/kg. So we thought that a dose of 1 mg/kg was sufficient. Next, we attempted a pharmacodynamic evaluation of a lower effective dose of 0.5 mg/kg. The results showed that the 0.5 mg/kg dose was less effective than 1 mg/kg. Therefore, we infer that 1 mg/kg is the best effective dose. Next, we did a pharmacokinetic study at the 1 mg/kg dose. The result showed that the concentration of free antibody in lung tissue was as low as approximate 0.1 μ g/mL on 7th day after LF2 administration, which was just the initial effective dose for *in vitro* cell function experiments. Therefore, the PK data below also support the 1 mg/kg is the optimal dose with 7-days interval.

AbL and LF2L are both the abbreviations for the group treated with low-dose LF2 monoclonal antibody (0.5 mg/kg), while AbH and LF2H both refer to the group treated with high-dose LF2

monoclonal antibody (1 mg/kg). In order to avoid confusion, the writing in the legend and in the figure have been unified by LF2L and LF2H. The definition has also been added to the legend and figures as you suggested. The isotope and species of LF1, LF2 and the control IgG has been added to the legend.

3) Since *Pear1* is a membrane protein, the effects seem in the mouse study could be due to depletion of *Pear1*+ cells via ADCC/ADCP or the direct functional effect as suggested by the authors. To support the claim that *Pear1* agonism is the relevant MOA, the authors should show data with an effectorless Fc or some other additional data.

Response: Thanks for your professional comments. IgG has four subclasses, named IgG1, IgG2, IgG3 and IgG4. Although all the subclasses have more than 90% identity on the amino acid level, each subclass however has a unique profile with respect to the length of hinge region, the number of inter-chain disulfide bonds, and Fc-effector functions¹.

In mouse IgG, the IgG2a and IgG2b isotypes were described to induce potent Fc-mediated effector functions in the murine system². The hierarchy of Fc-effector functions for the IgG subclasses in these immune functions was thus IgG2a>IgG2b>IgG1>IgG3. The LF1 monoclonal antibody which targeted mouse PEAR1 was mouse IgG1 isotype, which had a weak Fc-effector activity.

Among human IgG, IgG1 and IgG4 are the two most used subtypes. Because IgG1 has a high affinity for FcγRs, the IgG1 subtype is widely used for antibodies targeting tumor cells, which can produce strong ADCC and/or ADCP activities on tumor cells. Since human IgG4 only has a high affinity to FcγRI but weak affinity to all other FcγRs and thus will not have a detectable Fc-mediated effector function³. IgG4 subtype was widely used for antibodies acting on non-target cells, such as the PD-1 monoclonal antibody which acts on T cells and enhances the killing of tumor cells by T cells. The original isotype of LF2 monoclonal antibody which targeted human PEAR1 immunized from BALB/c mice was mouse IgG2b. Then, in order to reduce the immunogenicity of the antibody in human, we performed humanization of the antibody. In order to avoid the platelet clearance induced by antibody-mediated ADCC/ADCP effect and resulting in serious side effects, the human IgG4 subtype was selected. And the S228P mutation was introduced into the Fc region to prevent the fab change and stabilize the IgG4 molecule⁴. The antibody used in **Fig. 4b-d** was just the humanized LF2 monoclonal antibody, in which the Fc was human IgG4 and had a weak Fc-effector activity. Moreover, the F(ab)₂ of LF2 in which the Fc has been deleted was obtained by pepsase digestion. Considering the short lifespan of Fc-deficiency mab *in vivo*, the function of F(ab)₂ was detected in cultured PF fibroblasts. The data showed that F(ab)₂ could inhibit the synthesis of ECM (figure blow), indicating that the function of LF2 in fibroblast is not dependent on Fc. Although the ADCC/ADCP cannot be excluded completely, it can be concluded that the function of LF1 and LF2

in vivo was mainly dependent on PEAR1 agonism. The isotopes of LF1 and LF2 have been added to the figure legends.

4. Manuscript needs grammar editing.

Response: As you suggested, the grammar has been edited in this new version.

To Reviewer #2:

1. While platelet function does not appear to be affected (Supp Fig 10) since PEAR1 signalling is activated upon platelet aggregation and independent of activation, this suggests that platelets may be directly involved in the progression of Bleo pulmonary Fibrosis and this aspect of platelet biology was earlier reported by Piguet in 1994. Authors should consider these earlier publications and incorporate their current findings as to the direct involvement of non-activated but aggregated platelets in the progression or resolution of fibrosis.

1) Piguet PF and Vesin C. Pulmonary platelet trapping induced by bleomycin: correlation with fibrosis and involvement of the beta 2 integrins. *Int J Exp Pathol* 1994;75(5):321-8.

2) Piguet PF, Tacchini-Cottier F, Vesin C. Administration of anti-TNF-alpha or anti-CD11a antibodies to normal adult mice decreases lung and bone collagen content: evidence for an effect on platelet consumption. *Am J Respir Cell Mol Biol.* 1995;12(2):227-31.

3) Piguet PF, Van G, Guo J. Heparin attenuates bleomycin but not silica-induced pulmonary fibrosis in mice: possible relationship with involvement of myofibroblasts in bleomycin, and fibroblasts in silica-induced fibrosis. *Int J Exp Pathol* 1996;77(4):155-161.

Response: Thanks for your professional comments. The previous findings reported by Piguet showed that bleomycin induced pulmonary fibrosis is correlated with platelet trapping and anti-CD11a antibody decreased the platelet trapping and collagen deposition, which indicated that platelet might play a role in the progression of Bleo-induced pulmonary fibrosis. Although the mechanism remains unclear. And the previous findings showed that PEAR1 is activated upon platelet-platelet contact and independent of platelet activation⁵. Although the function of PEAR1 in platelets remains controversial. FcεR1α, dextran sulfate and fucoidan potentiated platelet aggregation through direct binding to EGF-like 13 domain of PEAR1 in human platelets. However, Criel, M., et al⁶ and our group showed that the *Pear1* deficiency does not affect the murine platelet function. Given the study of Piguet, PEAR1 may regulate fibrosis progression by affecting platelet properties other than activation. Therefore, it was not sufficient to detect platelet activation alone indeed. In order to reveal the role of platelet PEAR1 in Bleo-induced pulmonary fibrosis, the platelet-specific *Pear1* deficiency mice (*PF4-Cre⁺Pear1^{fl/fl}*) have been constructed. The results in

Supplementary Fig. 13h-k showed that there was no significant difference of lung function or collagen deposition between $PF4-Cre^+Pear1^{fl/fl}$ and the control mice ($Pear1^{fl/fl}$), indicating that platelet PEAR1 was not involved in the progression of Bleo-induced pulmonary fibrosis. The function of PEAR1 in pulmonary fibrosis is mediated mainly by fibroblasts.

2. There does not appear to be any data on the effect of the mAb raised against the humanized PEAR1 on human fibroblasts that are readily accessible. Moreover, one may expect to see differences in response when fibroblasts from established fibrotic lungs are compared to normal “control” human lung fibroblasts. Since most patients presenting with PF have established fibrotic tissue, whether the cells in the fibrotic lung would respond to the antibody exposure would be an important issue.

Response: As you suggested, HFL1 which is a fibroblast cell line that was isolated from the lung of a white, normal embryo (ATCC; CCL-153) and PF which are primary Lung fibroblasts isolated from the lung of a pulmonary fibrosis patient (ATCC; PCS-201-020) were used to detect the function of LF2 mab *in vitro*. The data in **Supplementary Fig. 12** showed that the synthesis of ECM-related gene such as Postn, Timp1, FN1 in both HLF1 and PF fibroblasts in response to TGF- β were significantly inhibited by LF2. These data demonstrated that fibroblasts from established fibrotic lungs could respond to the LF2 antibody, which was critical for the application of this antibody in the treatment of pulmonary fibrotic diseases. Thanks for your suggestion.

3. Do the authors have any data regarding the penetration by the LF1 antibody of epithelial cells or cell junctions to enter the interstitial tissue to access the fibroblasts and myofibroblasts present there?? Is the antibody acting at the epithelial cell to inhibit EMT? Does the antibody inhibit myofibroblast differentiation from resident fibroblasts (after stimulation with factors such as TGFb)?

Response: In the lung, alveolar epithelial cells and capillary membrane form a barrier that plays a central role in gas exchange, permeability regulation, fluid clearance, and host defense. However, the alveolar-capillary barrier is damaged during acute lung injury⁷. Pulmonary fibrosis is a pathological change caused by repeated injuries and repair dysfunction of the alveolar epithelium. The alveolar barrier is impaired in IPF lungs⁸. As you suggested, the distribution of LF2 antibody was detected in bleomycin-induced fibrosis lung. The podoplanin (PDPN) were co-stained to indicate the epithelial cells. The data in **Supplementary Fig. 11d** showed that the LF2 could enter the interstitium and distributed evenly in the lung.

The Flow cytometry data of mouse lung tissue and MLE12 cells showed there was no expression of PEAR1 on epithelial cells (the Figure a,b below). Therefore, we speculate that this antibody has no effect on epithelial cells.

The data in **Supplementary Fig. 12b** showed that the expression of α-SMA mRNA was inhibited by LF2 antibody. However, it cannot be concluded that the antibody inhibits the myofibroblast differentiation from resident fibroblasts, since myofibroblast should be identified by several markers together with α-SMA. α-SMA is also expressed in pericytes-derived Fibromyocytes (Cluster 4), Sftpc/Aspn high stromal cells (Cluster 0), Cyp2e1 high stromal cells (Cluster 3) and Activated fibroblasts (Cluster 7). The Single cell RNA-seq analysis of pulmonary FB showed that *Pear1* deficiency did not affect the proportion of myofibroblasts, but enhanced the expansion of Activated-FB cluster which was enrich of marker genes involved in ECM development and appeared only when induced by bleomycin. Consequently, the antibody inhibits the transformation into “activated” fibroblasts, not myofibroblasts from resident fibroblasts.

4. What does PEAR1 deficiency in aging mice do to staining for α SMA and detection of myofibroblasts?

Response: Staining for α SMA was performed in aged mice as you suggested. The data showed that the α -SMA⁺ fibroblasts were more in the lungs of *Pearl1*^{-/-} aged mice compared with the *WT* aged mice (Figure below). However, as the statement in the response to Question 3, the α -SMA⁺ cells cannot be equivalent to myofibroblasts.

5. In relation to the early work of Piguet and colleagues, is this involvement of PEAR1 exclusive to bleomycin treatment (especially relevant to ref 3 by Piguet)? Many drugs showing effects in bleomycin induced experimental fibrosis have failed in moving into human studies.

Response: As you suggested, the roles of PEAR1 in fibrosis were evaluated by another pulmonary fibrosis model induced by amiodarone. Amiodarone, a cardiac antiarrhythmic agent, has been associated with the development of interstitial pulmonary fibrosis in patients receiving prolonged therapy with the drug. Amiodarone-treated lungs developed increased interstitial thickening and marked deposition of collagen, which morphologically resemble IPF in humans. The results showed that *Pearl1* deficiency significantly aggravated collagen accumulation and LF2 suppressed the deposition of ECM induced by amiodarone (Supplementary Fig. 11e, f), indicating that the role of PEAR1 is reproducible in different fibrosis models and maybe a promising target for pulmonary fibrosis therapy.

To Reviewer #3:

1. The authors may improve the presentation of their manuscript by using NPG Language Editing service.

For example: “IPF has a poor prognosis, is unpredictable, leads to irreversible progression and eventually respiratory failure and death”

“It was showed that Pear1 is also expressed in endothelial cells”

“Moreover, isolated and cultured FBs from WT lung tissue were almost all expression of Pear1”

“Observation of aged mice, we found that.....”

Response: Thank you for your suggestion. The manuscript has been edited by a native English speaker.

2. It is confusing why monoclonal antibody targeting human Pear1 actually activated pear1. More explanation is needed.

Response: Thank you for your comments. *Pear1* deficiency in lung fibroblast enhanced the expression of ECM genes. While the LF2 monoclonal antibody inhibited the expression of ECM genes, which demonstrate an opposite role of LF2 with *Pear1* deficiency. Therefore, we speculate that LF2 is an activating antibody of PEAR1. In spite of classical mechanism of actions exerted by monoclonal antibody, such as neutralization, blocking, CDC, ADCC, therapeutic antibody could also act as agonists of the receptors, essentially replacing the activity of the normal ligand. The agonist activity can occur when the antibody binds the receptor in a manner that mimics the binding of the physiological ligand resulting in antibody-mediated agonism. The high-affinity immunoglobulin E receptor α (Fc ϵ R1 α) has been identified as a natural ligand for human PEAR1, and Fc ϵ R1 α potentiated platelet aggregation and led to PEAR1 phosphorylation through direct binding to epidermal like growth factor (EGF)-like repeat 13 of PEAR1 in human platelets⁹. One previous study by Kardeby *et al* reported that natural fucose-based polysaccharides and synthetic glycopolymers stimulate PEAR1 by direct binding to EGF-like 13 domain of PEAR1¹⁰. Thus, EGF 13 may be an important domain mediated PEAR1 activation, and we suspect the epitope of the

antibody which activates PEAR1 may be in or near this region. However, the detailed mechanism requires further study.

3. *Biological replicates and batch information in the single cell mRNA-seq experiment should be presented.*

Response: Lung biopsies derived from 5 mice per group were pooled for tissue digestion and single cell isolation. All of the single cell samples were processed in the same batch for tissue collection, flow cytometry sorting, single cell isolation and library preparation, and sequencing. There are no technical batch issues involved in the procedure. The detailed information has been supplemented in the method as you suggested. Thank you for your suggestion.

4. *Sftpc and Cyp2e1 are markers of lung epithelial cells. However, the authors labelled cluster 3 and cluster 0 as stromal cells? Any contamination?*

Response: Thanks for your comments. Single cells were sorted by flow cytometry to remove hematopoietic cells (CD45⁺), endothelial cells (CD31⁺), epithelial cells (CD326⁺). Only CD326⁻CD45⁻CD31⁻ cells are subjected to scRNA-seq. Because sorting leak is unavoidable, we further removed cell clusters that are positive in either one of the three markers during data analysis (**Supplementary Fig. 3**). Therefore, all cell clusters in Fig. 2 are the purified mesenchymal cells. As shown in the figure below, all cell clusters including Cluster 0 & 3 are negative for the epithelial marker Epcam (CD326). Therefore, there is no contamination from epithelial cells.

The alveolar surfactant protein markers *Sftpc* and *Scgb1a1* are positive in Cluster 0 (**Fig. 2e**). Cluster 0 is also positive in *Aspn* (**Fig. 2e**) and *Acta2* (**Extended Fig. 6e**), displaying certain myofibroblasts feature. Therefore, Cluster 0 is likely alveolar FB and/or adventitial FB.

Cluster 3 is positive in *Cyp2e1* and *Lhfpl2* (**Fig. 2e**). According to Human Protein Atlas, *Lhfpl2* is strong in macrophage (c-0, c-2), followed by Fibroblasts (c-5), but weak or absent in Type I or II alveolar cells (c-1, c-6, c-10)---

<https://www.proteinatlas.org/ENSG00000145685-LHFPL2/single+cell+type/lung>

Similarly, *Cyp2e1* expression in fibroblasts (c-5) is higher than or similar as that in alveolar epithelial cells (c-1, c-6, c-10)

<https://www.proteinatlas.org/ENSG00000130649-CYP2E1/single+cell+type/lung>

5. Based on Figure 2a, Cluster 6 includes two separate groups. Why are they defined as one cluster?

Response: Thanks for your comments. We have revised our manuscript by addition Extended Fig. 7 to address this issue.

We performed sub-clustering on Cluster 6, generating two major subclusters 0 and 1 (**Supplementary Fig. 7a**). Subcluster-0 is positive in multiple SMC markers, including Acta2, Tagln, Mhy11, Cnn1, Agtg2, and also Mcam, therefore is a cluster of airway SMC (**Supplementary Fig. 7b**). The subcluster-1 demonstrated positive expression in multiple lipofibroblasts markers, including Col1a2, Aspn, Apoe, Plin2 (**Supplementary Fig. 7c**)¹¹. The two subclusters of Cluster 6 demonstrated little difference between KO.Ble and WT.Ble (**Supplementary Fig. 7a**), and thus were not a focus of current study.

6. Figure 3g: PP1 α Ab IP generates more Pear1 signal than Pear1 Ab IP. Why?

Response: Thank you very much for your careful review. As you pointed out, when using PP1 α Ab IP, the PEAR1 signal was stronger than PEAR1 Ab IP. We speculate that this may be due to the low enrichment efficiency of the PEAR1 Ab we originally used, and the band of PEAR1 which was pulled down by this antibody was weak. Therefore, several kinds of other antibodies have been tried. It can be seen from the new Fig. 3g that PEAR1 signal which was pulled down by the new antibody was much stronger. In fact, it is not suitable to compare the signals of the two channels, because different antibody is used for IP, and the enrichment efficiency of the antibodies is not the same. In order to avoid confusion, the two sets of IPs have been separated into two different figures.

7. How specific is Pear1 antibody? The left side of hPEAR1 signal in Supplementary Fig.9b looks a bit different from the other areas. In addition, the voltage is quite high for PEAR1 signal in Supplementary Fig.9c. Is it because of a problem with the antibody?

Response: Thank you very much for your careful review. As you pointed out, the left side of hPEAR1 signal in Supplementary Fig.9b looks overexposed, and the figure has been replaced by a shorter exposure image. Since the expression of PEAR1 on the surface of normal fibroblasts is not high, a high voltage is required in flow cytometry to allow all PEAR1-positive cells to be detected. The data in **Supplementary Fig.9c** showed that human PEAR1 signal can only be detected on fibroblasts from humanized *Pear1* mice, but not on fibroblasts from *WT* mice, while the mouse PEAR1 signal was detected on fibroblasts from *WT* mice, but not on fibroblasts from humanized *Pear1* mice. These two PEAR1 antibodies can distinguish human and mouse PEAR1, indicating that these two PEAR1 antibodies are specific.

To Reviewer #4

1.Line 24 and next: In the introduction the authors state that Pear1 may activate platelets by binding its EMI domain to an unknown ligand. This is not entirely correct: a putative ligand has been identified, the high affinity immunoglobulin E receptor subunit alpha (FcεR1α), present on human platelets, and involved in human platelet activation amplification via PEAR1- FcεR1α interactions. On murine platelets, both Pear1 and especially FcεR1α are expressed in lower amounts than on human platelets, strongly reducing contributions by Pear1- FcεR1α interactions to platelet activation, to insignificant levels. So, the statement in line 24 about “Pear1” would be "partly" correct for human PEAR1. Murine Pear1 deficiency does not affect platelet function, induced by regular platelet agonists, but semi-specific Pear1 agonists, such as dextran sulfate and fucoidan can potently activate mouse platelets, in a Pear1-dependent manner. It is surely correct that FcεR1α may not be the primary or major receptor and other ligands or receptors may be identified in the future. So, these phrases might be slightly adapted (murine Pear1 vs. human PEAR1).

Response: Thank you for your suggestion. As you suggested, the description of PEAR1 in the introduction has been adapted more specifically to human or mouse PEAR1. And the statement of the ligands for PEAR1 which has been identified has also been added into the introduction. The statement has been revised as “The high-affinity immunoglobulin E receptor α (Fc ϵ R1 α) has been identified as a natural ligand for human PEAR1, and Fc ϵ R1 α potentiated platelet aggregation and led to PEAR1 phosphorylation through direct binding to epidermal like growth factor (EGF)-like repeat 13 of PEAR1 in human platelets. PEAR1 was also reported to be a signaling receptor for dextran sulfate and fucoidan in human, but not in mouse platelets. However, a study showed that the *Pear1* deficiency does not affect the murine platelet function”.

2. The present study specifies mesenchymal cell proliferation but it is not clear whether this study describes a proliferation or rather a differentiation study, or the mixture of both. In megakaryocyte progenitors and endothelial cells, PEAR1 gene-deficiency enhances cellular proliferation in vitro and in vivo, which has been coupled to increased Akt-phosphorylation and reduced PTEN expression. In those studies, the reduced expression of PTEN in Pear1-/- mice has been advocated to explain the proliferative phenotype. As is also found in the present work, Pear1 does not play a role in developmental angiogenesis, but in models of neoangiogenesis and wound healing, Pear1 plays a clear role. Did the authors in the present work investigate cellular proliferation in their lung sections (use of cellular proliferation markers, such as PCNA or Ki67)

and did they look into Akt-P levels during mouse aging and/or during bleomycin treatment? Did they do co-staining with the epithelial, mesenchymal or fibroblast markers? That will determine whether mesenchymal cell proliferation is the driver or not. In the extended part, several lung sections are stained for specific markers. It would be very informative to combine some of this staining with examples where a proliferation marker would be used, such as PCNA or Ki67 to determine whether the cellular differentiation, presently reported coincides or not with increased proliferation of mesenchymal cells, or fibroblasts. That will clarify whether the increased deposition of ECM components results from a transdifferentiating effect or a combined proliferative and/or functional effect.

Response: Thanks for your professional comments. As you suggested, the ki67 staining was performed in lung sections from *WT* and *Pear1*^{-/-} mice induced by bleomycin. The podoplanin (PDPN), Pdgfr was co-stained to indicate epithelial cells and fibroblasts, respectively. The data showed that the ki67⁺ fibroblasts in lung sections from *Pear1*^{-/-} mice were more than that from *WT* mice, while there was not a significant difference of the number of ki67⁺ epithelial cells between lung sections from *WT* and *Pear1*^{-/-} mice (**Supplementary Fig. 1i, j**). Furtherly, the proliferation of the primary fibroblasts isolated from *WT* and *Pear1*^{-/-} mice was evaluated by CCK8 assay. The data showed that *Pear1* deficiency enhanced the proliferation of fibroblasts (**Supplementary Fig. 1h**). Moreover, the data in **Supplementary Fig. 12b** demonstrate a role of PEAR1 in increasing the expression of ECM genes in fibroblasts. These data indicated that PEAR1 was involved in both proliferation and activation of fibroblasts.

The flow cytometry assay of fresh lungs, IF staining of fibroblast marker Pdgfra in frozen lung sections showed a significant high number of fibroblasts in the lung from *Pear1*^{-/-} mice compared with that from *WT* mice in response to bleomycin. The sc-RNA seq and bulk RNA seq assay showed there was much more activated fibroblasts in the lung from *Pear1*^{-/-} mice compared with that from *WT* mice in response to bleomycin. Therefore, increased deposition of ECM components in *Pear1*^{-/-} lungs should be the result of the combined effect of proliferation and transdifferentiating to activated fibroblast.

3. Interestingly, in this study, the authors found an association between Pear1 and Protein Phosphatase 1, but did they do (semi)-quantitative analysis of its presence in lung sections of WT and Pear1^{-/-} mice? Or staining/western blots? Does its concentration shift during aging or bleomycin treatment? The authors now invoke phosphorylation of several growth factors to explain the cellular differentiation process, but can they exclude that differential Protein Phosphatase 1 in WT and Pear1^{-/-} mouse lungs are responsible for differential states of baseline Akt phosphorylation in both types of lung, driving cellular reorientation differently, as a result of different metabolism of the differentiated fibroblasts? PTEN has been reported to play a

crucial role in endothelial cells in angiogenesis, but it is not impossible that in mesenchymal cells/fibroblasts, a similar role exists for the PP1 α subunit, especially since PTEN is not picked up in the RNA sequencing. Since Protein Phosphatase associates with Pear1, what is its fate in the Pear1 K.O. lung cells?

Response: The expression of PP1 α in cultured WT and Pear1^{-/-} fibroblast was detected by western blotting. The data showed that there was no significant difference of PP1 α expression in WT and Pear1^{-/-} fibroblast, indicating that Pear1 deficiency did not affect the expression of PP1 α (Supplementary Fig. 9f).

The expression of PTEN in cultured WT and Pear1^{-/-} fibroblast was also detected. The data showed that there was no significant difference of PTEN in WT and Pear1^{-/-} fibroblasts, indicating that PTEN deficiency did not affect the expression of PTEN. Moreover, unlike PP1 α , PTEN did not directly bind PEAR1 in lung fibroblasts (Supplementary Fig. 9g).

PEAR1 did not affect the expression of PP1 α , however, PEAR1 interacted with PP1 α and regulated the function of PP1 α in fibroblasts. The data in Fig. 4 f-g showed that the levels of p-smad2/3, p-AKT, p-erk1/2 and p-JNK were enhanced in Pear1^{-/-} fibroblasts, indicating that the activity of PP1 α may be inhibited in Pear1 KO fibroblasts which leads to higher levels of phosphorylation.

4. In platelets and endothelial cells, PEAR1 activation with an antibody triggers rapid PEAR1 phosphorylation and activation of Akt and all of its coupled pathways. (These will differ from cell to cell.) Yet, this action is a cell signaling response to an external activation signal, meaning that this is transient. In contrast, in PEAR1 gene-deficient cells, there is a permanent impact of the gene-deficiency on several pathways and on gene-regulation and protein expression. It is difficult to understand how the instillation of an activating anti-Pear1 antibody can control the “same” cellular differentiation cascade as found during gene-deficiency. The antibody is given with an interval of one week. How long does the effect of the antibody last, as measured e.g. by PEAR1 or Akt phosphorylation or phosphorylation of any other signaling molecules, or Protein Phosphatase 1? The results are clear: there is a correction of the phenotype, but is this the result of a signaling cascade triggered by the antibody that is controlling the same pathways as those that are dysregulated during bleomycin treatment? Are these the same pathways, as those that are affected in the K.O. cells?

Response: Thanks for your professional comments. The bulk tissue RNA-seq analysis data showed that although there was a slight difference in the gene expression levels of fibroblasts isolated from

normal *WT* and *Pear1*^{-/-} mice, the expression levels of fibrosis-related genes have no significant difference between normal *WT* and *Pear1*^{-/-} mice. In response to bleomycin, the expression levels of fibrosis-related genes were significantly increased, and *Pear1* deficiency led to a more significant increase in the expression levels of fibrosis-related genes than *WT*. Thus, the inhibition of PEAR1 in the fibrosis-related genes is dependent on fibrotic stimulus-mediated signaling pathway, indicating that PEAR1 may be activated as feedback by an unknown mechanism and restrain stimulus-mediated signaling pathway. LF2, as an activating antibody of PEAR1, inhibit the expression levels of fibrosis-related genes and the progression of fibrosis, indicating that PEAR1 cannot be activated completely for an uncertain cause. Maybe the ligand of PEAR1 is lack in fibrosis progression, which requires further studies.

Considering that fibroblasts make up only a tiny fraction of the lung tissue, the detection of the signaling pathways in entire lung tissue will be susceptible to interference by other cells. Therefore, the function of LF2 in human lung fibroblasts was detected *in vitro*. The results showed that LF2 inhibited the expression of ECM genes (Supplementary Fig. 12), indicating that LF2 controls the same pathways as those are dysregulated in KO cells.

Monoclonal antibodies have the characteristics of targeting specifically and long half-life *in vivo*, and the dosing interval usually supports more than 7 days. The lifespan of LF2 was studied in human *Pear1* transgenic mice at a dose of 1 mg/kg. The data showed that the concentration of free antibody in the lungs was >0.1 µg/mL on the 7th day after LF2 administration. According to the cell function assay *in vitro*, this concentration can still achieve the effect of activating fibroblasts and inhibiting the expression of cell matrix-related genes. Thus, pharmacokinetic data also supports the frequency of dosing every 7 days.

5. The aging phenotype is investigated in 12 months old mice. How old were the mice subjected to bleomycin treatment? Same question for the *Col1a2-CreERPear1^{ff}* mice? Is the age the same in all experiments? Is there a gender dependence, or is the same sex used throughout?

Response: The mice subjected to bleomycin treatment are usually 8~12 weeks old, containing *WT*, *Pear1*^{-/-}, humanized *Pear1* mice, *Col1a2-Cre⁺ER⁺Pear1^{ff}* mice and *PF4-Cre⁺Pear1^{ff}* mice. Since IPF is a disease with a male predominance, the mice used in all experiments are male. The detailed information has been added to the method.

6. Already in the abstract, the authors state “Mesenchyme-specific *Pear1* deficiency aggravated bleomycin-induced PF, confirming that *Pear1* modulates PF progression potentially via regulation of FB function”. This phrase leads to some confusion, because the notion of mesenchymal cell function and fibroblast function are mixed. The authors should provide their definition of a mesenchymal cell in the Supplementary part, supported (and explained) by the

corresponding flow cytometric definition. Flow cytometric definition is given in the legend of Figure 1: epithelial cells are defined as “CD326+CD45-CD31-“ and fibroblasts as “Pdgfra+CD45-CD31-“. Mesenchymal cells are defined as “Pdgfra+CD31-CD45-CD326-“. However, in the Results page 4, line 4, mesenchymal cells are defined as “Pdgfra+CD31-CD45-“, i.e. irrespective of their CD326 presence. In some places, mesenchymal cells are defined as Pdgfra+ only. Some clarification about the relation between lung mesenchymal cells, epithelial cells and fibroblasts, as detected in the present study by flow cytometry will avoid confusion and will make clear whether (activated) fibroblasts are meant or mesenchymal cells. Along these lines it would be informative to show in Fig 1f the difference for WT CD31-CD45- and Pear1-/- CD31-CD45- at the representative age used (see question 5) and 12 months of age. Encircle the relevant cell populations and report their proportions and distribution differences, using the standardized definitions.

Response: Thanks for your careful review. In order to avoid confusion, the flow cytometric definition in the legend of Fig. 1 has been revised as “The proportion of leukocytes (CD45+), endothelial cells (CD31+), epithelial cells (CD326+) and Pdgfra+ mesenchymal cells (CD45-CD31-CD326-) were analyzed in the lung tissue of Pear1-/- mice with or without treatment of bleo”. The cell populations have also been circled on the scatter plot in Fig. 1f. And the definition has been unified in the manuscript. Mesenchymal cells population is defined by CD45-CD31-CD326- and Pdgfra+ mesenchymal cells was a main population of mesenchymal cells. The sc-RNA seq analysis data showed that Pdgfra was expressed on almost all resident matrix FB and Activated FB. Therefore, Pdgfra+ cells were selected by flow cytometry for RNA-seq assay. The proportion of CD31-CD45- cells in lungs from WT and Pear1-/- mice with or without bleomycin treatment at 8~12 weeks and WT and Pear1-/- mice at 12 months of age has been shown in Supplementary Fig. 1e, f.

7. As mentioned above, it is important to distinguish between a model in which the increased PF is caused by enhanced proliferation of mesenchymal cells, subsequently translating in more fibroblasts and more deposition of EC components and a model in which activity dysregulation in mesenchymal cells will translate in more transformation into “activated” fibroblasts. The authors show that Protein Phosphatase associates with Pear1. How sure is it that this phosphatase suppresses fibrotic factors-induced intracellular signaling, as this is shown for TGF Beta, PDGF

and FGF? Fig 3f shows no differences between WT and KO bands after stimulation. Also, there appears to be little difference between WT and K.O. controls. Yet, Akt-P in Ctrl KO and after FGF appear to be comparable and higher than in the WT condition. Is this correct? In that case, this analysis would be parallel to what is seen in endothelial cells, where the knock-down or knock-out raises the baseline phosphorylation of Akt, thus generating a proliferative phenotype. It would be interesting to have some measurement of Protein Phosphatase 1 in WT vs KO fibroblasts, maybe with a similar analysis for PTEN (maybe not when the authors can confirm from their RNA sequencing data that it does not play a role in the present report). In that case, the enhanced Akt-phosphorylation reflects a knock-out related genetic and biochemical change, in turn driving all the aspects related to the changed phenotype.

Response: As the response to Question 2, ki67 staining, CCK8 assay, sc-RNA seq and the *in vitro*-cell function data demonstrated that PEAR1 play mixed roles in both proliferation and transformation into “activated” fibroblasts.

The expression of PTEN and PP1 α in cultured WT and *Pear1*^{-/-} fibroblast was detected by western blotting as you suggested. The data showed that there was no significant difference of PTEN or PP1 α in WT and *Pear1*^{-/-} fibroblast, indicating that PEAR1 did not affect the expression of PTEN or PP1 α . Moreover, unlike PP1 α , PEAR1 cannot directly bind PTEN (**Supplementary Fig. 9f, g**). These data together exclude the possibility that PEAR1 regulate fibroblast function by PTEN.

Actually, the phosphorylation levels in cells are both regulated by the stimulus signaling and phosphatase. The PP1 inhibitor significantly enhanced the phosphorylation levels in fibroblasts, indicating that the activity of phosphatase is at a high level against the stimulation of the cytokines in the serum and sustains a relative lower phosphorylation level in cultured fibroblasts. Considering this, since PEAR1 acts as a regulator for PP1, the enhanced phosphorylation by PEAR1 deficiency without extra fibrosis factors was reasonable. In *Pear1*-deficient fibroblasts, although the phosphorylation level of each molecule was only slightly increased, the proteins that undergo phosphorylation changes are widespread, which is consistent with the fact that PP1 is a phosphatase which has a broad range of substrate exactly. While PTEN specifically regulates the PI3K-AKT signaling pathway.

8. The PP1 inhibitor has a very pronounced effect on the phosphorylated bands, which reflects the importance of this phosphatase. The evident question here is: does this inhibitor (in the adequate concentration-range) have the same effect for WT and KO cells, before and after

bleomycin treatment? Moreover, does it have an effect on the baseline phosphorylation of PEAR1 itself?

Response: As you suggested, the effect of PP1 inhibitor on cultured *WT* and *KO* fibroblasts was detected. The data showed that the levels of p-AKT, p-erk1/2, p-P38 and p-JNK were significantly enhanced by PP1 inhibitor in a dose-dependent manner. The baseline phosphorylation levels were slightly higher in *KO* fibroblasts than that in *WT* fibroblast, and the levels of p-AKT, p-erk1/2, p-P38 and p-JNK raised to the comparable levels in *WT* and *KO* fibroblasts in response to PP1 inhibitor. The data indicated that PP1 plays broad and important roles in regulating the phosphorylation levels in fibroblasts (**Supplementary Fig. 9e**). Bleomycin was used to treat *WT* and *KO* fibroblasts, however, the levels of p-AKT, p-erk1/2, p-P38 and p-JNK were not changed in response to bleomycin at various concentrations (data not shown). It may be because the actual role of bleomycin in inducing fibrosis is to cause epithelial cell damage and stimulate inflammatory cells, not directly activate fibroblasts.

It has been reported that PEAR1 can be phosphorylated at Tyr-925 and Ser-953/1029 during platelet aggregation⁵. Considering that PP1 is a Ser/Thr phosphatase, Ser/Thr phosphorylation of PEAR1 was mainly measured in response to PP1 inhibitor. Since there is a lack of antibodies specifically against the phosphorylation formation of PEAR1 currently, an IP-WB method was used, in which PEAR1 protein was immunoprecipitated by a PEAR1 antibody and western blotting with anti-Phospho-(Ser/Thr) antibody. The data below showed that the baseline phosphorylation levels of PEAR1 was quite low in fibroblasts and were not affected by PP1 inhibitor, indicating that PP1 was not involved in Ser/Thr phosphorylation of PEAR1.

9. The English language requires some attention and should be checked by a native English speaker.

Response: Thank you for your suggestion. The manuscript has been edited by a native English speaker.

References

1. Dekkers, G., *et al.* Affinity of human IgG subclasses to mouse Fc gamma receptors. *MAbs* **9**, 767-773 (2017).
2. Nimmerjahn, F. & Ravetch, J.V. Divergent immunoglobulin g subclass activity through selective Fc receptor binding. *Science* **310**, 1510-1512 (2005).
3. Yu, J.F., Song, Y.P. & Tian, W.Z. How to select IgG subclasses in developing anti-tumor therapeutic antibodies. *J Hematol Oncol* **13**(2020).
4. Silva, J.P., Vetterlein, O., Jose, J., Peters, S. & Kirby, H. The S228P mutation prevents in vivo and in vitro IgG4 Fab-arm exchange as demonstrated using a combination of novel quantitative immunoassays and physiological matrix preparation. *J Biol Chem* **290**, 5462-5469 (2015).
5. Nanda, N., *et al.* Platelet endothelial aggregation receptor 1 (PEAR1), a novel epidermal growth factor repeat-containing transmembrane receptor, participates in platelet contact-induced activation. *J Biol Chem* **280**, 24680-24689 (2005).
6. Criel, M., *et al.* Absence of Pear1 does not affect murine platelet function in vivo. *Thromb Res* **146**, 76-83 (2016).
7. Budinger, G.R. & Sznajder, J.I. The alveolar-epithelial barrier: a target for potential therapy. *Clin Chest Med* **27**, 655-669; abstract ix (2006).
8. Ohta, H., Chiba, S., Ebina, M., Furuse, M. & Nukiwa, T. Altered expression of tight junction molecules in alveolar septa in lung injury and fibrosis. *American journal of physiology. Lung cellular and molecular physiology* **302**, L193-205 (2012).
9. Sun, Y., *et al.* A Human Platelet Receptor Protein Microarray Identifies the High Affinity Immunoglobulin E Receptor Subunit alpha (FcepsilonR1alpha) as an Activating Platelet Endothelium Aggregation Receptor 1 (PEAR1) Ligand. *Molecular & cellular proteomics : MCP* **14**, 1265-1274 (2015).
10. Kardeby, C., *et al.* Synthetic glycopolymers and natural fucoidans cause human platelet aggregation via PEAR1 and GPIIb/alpha. *Blood advances* **3**, 275-287 (2019).
11. Travaglini, K.J., *et al.* A molecular cell atlas of the human lung from single-cell RNA sequencing. *Nature* **587**, 619-625 (2020).

REVIEWERS' COMMENTS

Reviewer #1 (Remarks to the Author):

The authors have adequately addressed all of my concerns.

Reviewer #2 (Remarks to the Author):

I am satisfied that the authors have considered and answered my previous questions and have diligently provided additional data to support their statements.

Reviewer #3 (Remarks to the Author):

I have no more comments.

Reviewer #4 (Remarks to the Author):

The revised manuscript has considerably improved and clearly shows a role for PEAR1 in mesenchymal cells/fibroblasts, which differs from previously reported mechanisms, described for other cells.

The authors have carried out additional experiments, which have answered all my questions. There are important aspects of mesenchymal cell activation and transdifferentiation, coupled to fibroblast activation, which have been dealt with in detail. I agree with the authors that further studies are warranted in this area, but in follow-up studies. I have no further remarks to make.